# GoR: A Unified and Extensible Generative Framework for Ordinal Regression

**Hongxu Ma**[1][*][†]**, Han Zhou**[4][*]**, Kai Tian**[1]**, Xuefeng Zhang**[2]**, Chunjie Chen**[2]**,**
**Han Li**[2]**, Jihong Guan**[3]**, Shuigeng Zhou**[1][‡]

[1]Shanghai Key Lab of Intelligent Information Processing, and
College of Computer Science and Artificial Intelligence,
Fudan University, Shanghai, China
[2]Kuaishou Technology, Beijing, China
[3]School of Computer Science and Technology,
Tongji University, Shanghai, China
[4]Shanghai University of Finance and Economics, Shanghai, China

## Abstract

Ordinal Regression (OR), which predicts the target values with inherent order, underpins a wide spectrum of applications within diverse domains. The intrinsic ordinal structure and non-stationary inter-class boundaries make OR fundamentally more challenging than conventional classification or regression. Existing approaches, predominantly based on Continuous Space Discretization (CSD), struggle to model these ordinal relationships, but are hampered by boundary ambiguity. Alternative rank-based methods, while effective, rely on implicit order dependencies and suffer from the rigidity of fixed binning. Inspired by the advances of generative language models, we propose **G**enerative **O**rdinal **R**egression (**GoR**), a novel generative paradigm that reframes OR as a sequential generation task. GoR autoregressively predicts ordinal segments until a dynamic ⟨EOS⟩, explicitly capturing ordinal dependencies while enabling adaptive resolution and interpretable step-wise refinement. To support this process, we theoretically establish a bias–variance decomposed error bound and propose the **Co**verage–**Di**stinctiveness Index (**CoDi**), a principled metric for vocabulary construction that balances quantization bias against statistical variance. The GoR framework is model-agnostic, ensuring broad compatibility with arbitrary task-specific architectures. Moreover, it can be seamlessly integrated with established optimization strategies for generative models at a negligible adaptation cost. Extensive experiments on 15 diverse ordinal regression benchmarks across five major domains demonstrate GoR's powerful generalization and consistent superiority over SOTA OR methods. The code is available at https://github.com/snailma0229/GoR.git.

## 1 Introduction

Ordinal Regression (OR), also referred to as ordinal classification, addresses the prediction tasks where the target categories (or values) exhibit inherent ordinal relationships. As shown in Fig. 1(a), this paradigm has broad applications across various domains such as computer vision (e.g., facial age estimation (Niu et al., 2016; Li et al., 2022), image aesthetic assessment (She et al., 2021; He et al., 2022)) and recommendation systems (e.g., watch time prediction (Sun et al., 2024; Zhao et al., 2024), lifetime value prediction (Drachen et al., 2018; Ma et al., 2018)). Unlike conventional multi-class classification (Feng & Ge, 2026; Feng et al.) and continuous regression, the fundamental challenge in OR lies in explicitly modeling two critical properties: (1) the inherent ordinal structure among output labels, and (2) the non-stationary nature of semantic boundaries between adjacent categories.

Previous OR works have predominantly relied on Continuous Space Discretization (CSD) (Wang et al., 2025), as illustrated in Fig. 1(b). This strategy quantizes the target output space, potentially continuous

---

[*]Both authors contributed equally to this research. [†]Work done during the internship at Kuaishou Technology.
[‡]Corresponding author: Shuigeng Zhou <sgzhou@fudan.edu.cn>

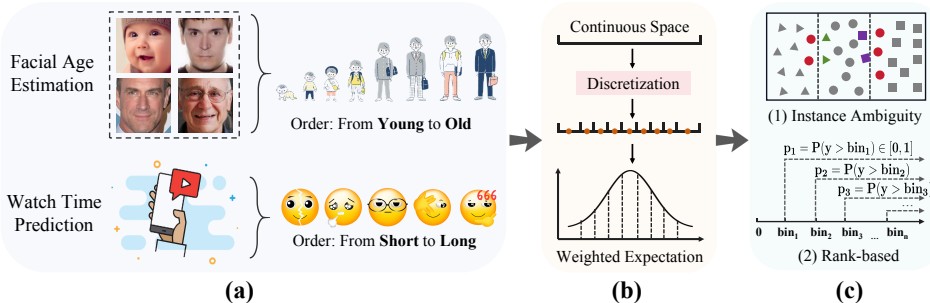

**(a)**    **(b)**    **(c)**

Figure 1: Overview of Universal Ordinal Regression. (a) Representative Ordinal Regression tasks with ordered labels. (b) The Continuous Space Discretization (CSD) workflow: discretizing continuous space into bins and using weighted expectation for prediction. (c) Two key research directions explored under the CSD framework.

or fine-grained ordinal, into a finite set of ordered discrete bins. The model typically outputs a softmax probability distribution over these bins, mapped to a prediction via probability-weighted expectation. Essentially, CSD simplifies learning by transforming the problem into a multi-class classification. Under this framework, subsequent research mainly explores in two directions.

As shown in Fig. 1(c), one is to tackle ambiguous inter-class boundaries by enhancing discrimination of boundary-proximal samples through reference comparisons (Li et al., 2021; Shin et al., 2022). However, its performance critically depends on efficient reference selection, which is often governed by unstable heuristics and limits the gains in wide-range scenarios where combinatorial reference points escalate selection complexity. Another is rank-based that implicitly encodes the ordinality via label transformation, reframing OR into sequential binary subtasks (Niu et al., 2016; Wang et al., 2023). Despite empirical success with theoretical guarantees (Chen et al., 2017), its order dependency resides solely in label definitions, leaving bin-wise predictions independent (See Proposition 1). Besides, the predefined discretization introduces rigidity: it amplifies head-category errors in long-tailed distributions, frequently seen in real-world tasks, and makes performance highly sensitive to bin granularity — wide intervals blur semantics, while narrow ones induce sparsity (Sun et al., 2024).

Inspired by the advances in generative language models (Liu et al., 2024; Liu et al.; Wang et al., 2026), we propose **G**enerative **O**rdinal **R**egression (**GoR**), a novel framework that reformulates ordinal regression as sequential token generation. GoR autoregressively predicts tokens representing ordinal value segments, whose cumulative summation yields the final prediction upon generating the $\langle EOS \rangle$ token. This design explicitly models sequential ordinal dependencies through conditional generation while enabling adaptive resolution via dynamic $\langle EOS \rangle$ prediction, circumventing the rigidity of fixed binning. For instance, in facial age estimation, the model may first predict a coarse token (50), then finer adjustments (+5, +3), culminating in the estimate (50+5+3=58 years). Each step progressively reduces prediction error by selecting tokens that provide an increasingly precise approximation. This step-wise refinement process mirrors human cognitive progression from coarse to precise estimation and implements successive approximation, offering interpretable intermediate predictions.

However, adapting this paradigm to general ordinal regression poses two key challenges. First, unlike purely compositional tokens in nature language process (NLP) tasks, GoR tokens uniquely encode dual semantics, *i.e.*, sequential ordinality and additive numerical relationships, necessitating specialized mechanisms to disentangle and exploit these intertwined properties. Since numerical values are infinitely decomposable and combinable, vocabulary design demands principled strategies to balance expressiveness and efficiency. Second, domain heterogeneity in ordinal label distributions, spanning diverse value ranges (e.g., $[0, 1]$ for quality scores vs. $[0, 100]$ for age estimation) and label granularity types (discrete vs. continuous), requires robust cross-domain generalization capabilities. Hence, effectively adapting the autoregression mechanism to universal OR tasks requires systematic vocabulary design and label decomposition strategies.

Through bias-variance decomposition, we derive a closed-form Mean Squared Error (MSE) bound that quantifies token selection trade-off between quantization bias and statistical variance, providing a theoretical foundation for vocabulary design. Building on this, we further propose the **Co**verage–**Di**stinctiveness Index (**CoDi**) to optimize token selection — maximizing coverage (bias

minimization) while suppressing common segments (variance reduction) — yielding a compact, recoverable vocabulary that enhances cross-task adaptability. Besides, as a model-agnostic framework, GoR's core component, i.e., the encoder-decoder architecture, permits flexible substitution with task-specific implementations and can be seamlessly integrated with existing optimization strategies (Bengio et al., 2009; Goodman et al., 2020; Shao et al., 2024) at a negligible adaptation cost, thereby ensuring its flexibility and extensibility for broader applications.

Our contributions are fourfold: (i) We expose the theoretical limitations of prevailing rank-based methods under the CSD paradigm and, in turn, propose the first generative formulation of ordinal regression as an autoregressive sequence generation task. (ii) We introduce GoR, a unified framework that models sequential ordinal dependencies via dynamic ⟨EOS⟩-terminated token generation, offering adaptive resolution and interpretable step-wise refinement. (iii) We establish a theoretical foundation based on MSE decomposition, accompanied by the Coverage–Distinctiveness Index (CoDi) to optimize the token vocabulary by bias-variance trade-off. (iv) We perform extensive experiments across 15 ordinal regression benchmarks spanning five domains, demonstrating GoR's strong generalization and consistent superiority over SOTA baselines.

The remainder of this paper is organized as follows. Section 2 formalizes the problem, theoretically analyzes rank-based methods' limitation, and introduces the proposed GoR framework, including its error-bound formulation, vocabulary construction, ordinal target sequencing, and encoder–decoder design. Section 3 reports extensive experiments and analyses across diverse domains. Section 4 concludes the paper. Due to space limit, we provide related work review in Appendix. B.

## 2 METHOD

### 2.1 PROBLEM DEFINITION

We consider the ordinal regression problem on a dataset $\mathcal{D} = \{(x_i, y_i)\}_{i=1}^N$, where $x_i$ denotes an input instance from heterogeneous modalities (e.g., visual data or multimodal embeddings) and $y_i \in \mathbb{R}_{\geq 0}$ represents its associated ordinal label. While conventional methods aim to learn a direct mapping $g(x_i) \approx y_i$, GoR reformulates this task as a sequence generation problem by establishing a bijection between the continuous label space and a discrete token sequence space. Specifically, each label $y_i$ is encoded into a variable-length sequence $\tau_i = \{\tau_i^t\}_{t=1}^{T_i}$ of length $T_i$, where each token $\tau_i^t$ is drawn from a predefined vocabulary $\Omega = \{\omega_j\}_{j=1}^V$ of size $V$. To close the loop between the discrete and continuous spaces, a sequence scoring function $r(\cdot)$ is defined to reconstruct the numerical label via an additive lookup table $\nu : \Omega \to \mathbb{R}$, such that $y_i \approx r(\tau_i) = \sum_{t=1}^{T_i} \nu(\tau_i^t)$. In practice, $\Omega$ is augmented with control tokens $\{\langle \text{SOS} \rangle, \langle \text{EOS} \rangle, \langle \text{PAD} \rangle\}$ for parallel sequence processing; however, as these tokens carry null semantic value in the label space ($\nu(\omega) = 0$), they are omitted from the formal mathematical derivations for clarity.

### 2.2 THEORETICAL ANALYSIS

Our theoretical analysis consists of two parts: characterizing the limitation of rank-based CSD approaches, and establishing GoR's theoretical foundation via MSE decomposition, which derives a closed-form MSE bound to guide vocabulary design.

**Limitation of rank-based methods.**

**Assumption 1.** *Define a sequence of binary random variables $\mathbf{B}_i^m = 1(y_i > c_m)$, where $1(\cdot)$ denotes the indicator function and $c_m$ is the right boundary of the $m$-th interval. Then $\mathbf{B}_i = \{\mathbf{B}_i^1, \ldots, \mathbf{B}_i^M\}$ constitutes a set of non-mutually exclusive binary decisions that together describe the position of $y_i$. Rank-based methods approximate the true conditional distribution $P_{true}(\mathbf{B}_i \mid \mathbf{x}_i) = \prod_{m=1}^M P(\mathbf{B}_i^m | \mathbf{B}_i^{<m}, \mathbf{x}_i)$ by assuming conditional independence across all binary decisions: $P_{naive}(\mathbf{B}_i | \mathbf{x}_i) = \prod_{m=1}^M P(\mathbf{B}_i^m | \mathbf{x}_i)$. Based on this factorization, the final prediction is obtained as $\hat{y}_i = \sum_{m=1}^M P(B_i^m = 1) \cdot (c_m - c_{m-1})$.*

Building on this conditional independence assumption, we quantify the resulting approximation error:

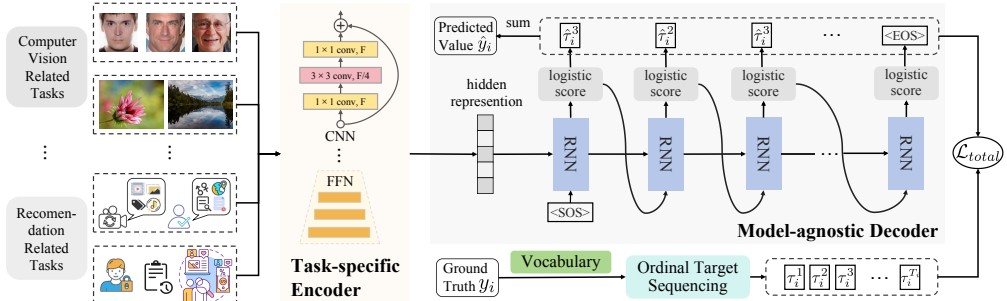

Figure 2: The framework of our proposed *Generative Ordinal Regression* (GoR), which adopts a flexible encoder-decoder architecture with the model-agnostic nature of both the encoder and decoder.

**Proposition 1** (Independence Limitation in Rank-based CSD methods).

$$D_{\text{KL}} \left( P_{true}(\mathbf{B}_i|\mathbf{x}_i) \| P_{naive}(\mathbf{B}_i|\mathbf{x}_i) \right) = \sum_{m=1}^{M} \mathbb{E}_{\mathbf{B}_i^{<m}}[D_{KL}^{(m)}], \tag{1}$$

where $D_{\text{KL}}^{(m)}$ measures the divergence between $P(\mathbf{B}_i^m|\mathbf{x}_i)$ and $P(\mathbf{B}_i^m|\mathbf{B}_i^{<m}, \mathbf{x}_i)$.

The complete derivations are provided in Appendix C. Proposition 1 reveals that naive discretization exhibits systematic modeling errors stemming from its inability to capture inter-interval dependencies. This phenomenon manifests as an approximation error quantified by cumulative KL divergence scaling with the conditional mutual information between adjacent intervals. To overcome this limitation, we recast ordinal regression through an autoregressive framework that explicitly captures sequential dependencies among latent tokens.

**MSE bound.** We denote $\nu(\tau_i^t)$ as a discrete random variable $C_i^t$, which takes values in $\nu(\omega_j)_{j=1}^V$. The model aims to approximate $C_i^t$ with its prediction $\hat{C}_i^t$. Let $B = \max_t |\mathbb{E}[\hat{C}_i^t \mid \theta] - C_i^t|$ denote the maximum per-step bias, and define the variance term $V_{var} = \max \mathbb{V}(C_i^t) \leq \frac{(\omega_{\max} - \omega_{\min})^2}{4}$, where $[\omega_{\min}, \omega_{\max}]$ is the range of $\{\omega_j\}_{j=1}^V$.

**Theorem 1** (Error Bound of Generative Ordinal Regression). *By a bias-variance decomposition (ignoring irreducible noise), the mean squared error of GoR satisfies:*

$$\mathbb{E}\left[(\hat{y}_i - y_i)^2\right] \leq T_i^2 B^2 + T_i^2 V_{var} \leq T_i^2 B^2 + T_i^2 \frac{(\omega_{max} - \omega_{min})^2}{4} \tag{2}$$

The detailed derivation and illustrations is in Appendix D. This error bound demonstrates that minimizing prediction error requires coordinated control of three critical factors: (i) token-sequence length $T_i$, (ii) maximum per-step bias $B$, and (iii) per-step variance $V_{var}$. Guided by these findings, we formulate three axiomatic principles for vocabulary design: (1) $\Omega$ must support approximation of all target values $\{y_i\}_{i=1}^N$ through finite unique tokens, ensuring bounded approximation bias. (2) Joint optimization of bias and variance via dual mechanisms—preventing sample imbalance bias through coverage constraint while suppressing variance via vocabulary sparsity control. (3) Parametric invariance across datasets, enforcing robustness to distribution shifts through scale-agnostic token contributions. These principles collectively ensure rigorous error control while maintaining practical applicability.

## 2.3 Framework

As illustrated in Fig. 2, GoR reformulates the regression task by employing a model-agnostic encoder-decoder architecture to learn the conditional probability distribution $P(\tau \mid x)$ over a discrete token sequence. This architecture comprises two key components: (1) a task-specific encoder for feature extraction, and (2) an architecture-agnostic autoregressive decoder for sequential prediction. The encoder adapts to arbitrary input modalities (e.g., text, images, or tabular), while the decoder

generalizes across sequence modeling paradigms — compatible with both RNN-based (Schuster & Paliwal, 1997; Chung et al., 2014) and Transformer-based architectures (Vaswani et al., 2017)[1].

### 2.3.1 PROBABILISTIC FORMULATION

Instead of directly regressing the scalar target $y_i$, GoR models the conditional distribution of its sequence representation $\tau_i$ given the latent feature $h_i$. By applying the probability chain rule, the joint probability of generating the complete sequence is factorized autoregressively as follows:

$$P_\theta(\tau_i \mid h_i) = P_\theta(\tau_i^1, ..., \tau_i^{T_i} \mid h_i) = \prod_{t=1}^{T_i} P_\theta(\tau_i^t \mid h_i, \hat{\tau}_i^{<t}) \tag{3}$$

where $\theta$ denotes the model parameters.

### 2.3.2 ARCHITECTURE

**Encoder.** The encoder is input modality-specific. Structured feature vectors employ Feedforward Network (FFN), while images use convolutional networks like ResNet (He et al., 2016) or ViT (Dosovitskiy et al., 2020). Formally, the encoder maps $x_i$ to a latent representation $h_i = \text{Encoder}(x_i) \in \mathbb{R}^{L \times D}$ where $D$ denotes the feature dimension and $L$ is the input length expected by the decoder.

**Decoder.** Conditioned on the latent representation $h_i$, the decoder recursively generates the token sequence $\hat{\tau}_i = (\hat{\tau}_i^1, \ldots, \hat{\tau}_i^{T_i})$. At each time step $t$, the model predicts the next token $\hat{\tau}_i^t$ is sampled as:

$$\hat{\tau}_i^t = \arg\max_{\omega \in \Omega} P_\theta(\omega | h_i, \hat{\tau}_i^{<t}) = \arg\max_{w \in \Omega} \text{Softmax}\big(f_\theta(h_i, \hat{\tau}_i^{<t})\big). \tag{4}$$

Here, $f_\theta$ outputs the unnormalized logits, while $P_\theta$ denotes the normalized probability distribution.

**Extensibility.** Beyond flexible substitution with task-specific implementations, GoR can integrate existing optimization strategies with negligible adaptation cost, including curriculum learning (Bengio et al., 2009), N-gram (Goodman et al., 2020), and reinforcement learning such as GRPO (Shao et al., 2024). The detailed discussion of these extensions is provided in Sec. 3.2.4.

## 2.4 TOKENIZATION

### 2.4.1 VOCABULARY

We initialize the vocabulary $\mathcal{W} = \{\omega_j\}_{j=1}^W$ through a quantile-based selection strategy, which iteratively selects tokens based on a fixed percentile of the remaining label values, and subtracts them from the exceeding labels until residuals are negligible (Details in Alg. 2 of Appendix E). This initialization ensures comprehensive coverage of the observed value distribution while introducing computational challenges due to excessive vocabulary size. We therefore develop a principled pruning strategy based on our proposed *Coverage-Distinctiveness Index* (CoDi) for tokens as follows:

$$\text{CoDi}_j = \underbrace{\left(\frac{1}{N}\sum_{i=1}^N \frac{\text{count}(\omega_j, \tau_i)}{T_i}\right)}_{\text{Coverage}} \cdot \underbrace{\log \frac{N}{|\{\, i \mid \omega_j \in \tau_i \}| + 1}}_{\text{Distinctiveness}} \tag{5}$$

Here $\text{count}(\omega_j, \tau_i)$ is the count of token $\omega_j$ in the sequence $\tau_i$ while $|\{\, i \mid \omega_j \in \tau_i \}|$ denotes the number of sequences containing $\omega_j$. In CoDi, the *Coverage* term measures token usage frequency, which affects approximation bias, while the *Distinctiveness* term evaluates token uniqueness, influencing model variance.

Based on CoDi, we design a top-down vocabulary pruning strategy in Alg. 1: starting with the initial vocabulary, we iteratively remove tokens with the lowest CoDi in the initial vocabulary $\mathcal{W}$. After each removal, the retained percentage $\beta$ and threshold $\epsilon$ are utilized to control vocabulary size while preserving representational fidelity, achieving a favorable trade-off between computational efficiency and modeling power. The refined vocabulary $\Omega$[2] then serves as the foundation for formalizing *Ordinal Target Sequencing*.

---

[1]To intuitively illustrate the generative process, Fig. 2 depicts an RNN-based decoder, which can be replaced with any other decoder architecture.

[2]Without loss of generality, the token indices in the vocabulary are also sorted in descending numerical order.

---

**Algorithm 1:** Vocabulary pruning with CoDi

---

**Input:** Label set $Y = \{y_i\}_{i=1}^N$; Sequence set $\{\tau_i\}_{i=1}^N$; Threshold $\epsilon$; Initial vocabulary
$\qquad \mathcal{W} = \{\omega_j\}_{j=1}^W$; Minimum percentage of initial vocabulary retained $\beta$
**Output:** Pruned vocabulary $\Omega$ with $err < \epsilon$

1   $\Omega \leftarrow \mathcal{W}$;
2   Compute CoDi for all $\omega_j \in \Omega$;
3   $err \leftarrow$ evaluate$(\Omega)$;
4   **while** $err < \epsilon$ **and** $|\Omega| \geq \beta|\mathcal{W}|$ **do**
5     $\omega^- \leftarrow \arg\min_{\omega_j \in \Omega} \text{CoDi}_j$;
6     $\Omega \leftarrow \Omega \setminus \{\omega^-\}$;
7     Update sequence set $\{\tau_i\}_{i=1}^N$ based on $\Omega$;
8     Update error metric $err \leftarrow \max\{\frac{y_i - r(\tau_i)}{y_i}\}_{i=1}^N$;
9   **end**
10   **return** $\Omega$;

---

### 2.4.2   Ordinal Target Sequencing

Ordinal target sequencing aims to encode each target $y_i$ into a token sequence $\tau_i$ by adhering to three core principles. First, for efficiency, the sequence length $T_i$ must be minimized to simplify the learning process. Second, for representational accuracy, the reconstruction error must be bounded by a relative tolerance $\epsilon$, i.e., $\frac{|y_i - r(\tau_i)|}{y_i} \leq \epsilon$. Finally, the sequence must enforce a coarse-to-fine monotonicity through a descending order of token values ($\tau_i^t \geq \tau_i^{t+1}$). To satisfy these principles, we develop a greedy decomposition algorithm that iteratively selects the largest admissible token $\tau_i^t \in \Omega$ from the residual value $\tau_i^t = \max\left\{w \in \Omega \mid \nu(w) \leq y_i - \sum_{k=1}^{t-1} \nu(\tau_i^k)\right\}$, terminating when the residual falls below $\epsilon$. This procedure guarantees: (i) minimal $T_i$ for given $\epsilon$ by design, (ii) monotonic token values ensured by the decomposition process, and (iii) $\mathcal{O}(|\Omega|)$ time complexity via pre-sorted vocabulary search. The resultant sequences provide compact yet precise representations while maintaining generation consistency across the dataset.

### 2.5   Training and Inference

**Training loss.** The primary objective is to minimize the negative likelihood:

$$\mathcal{L}_{NLL} = -\sum_{i=1}^N \log P_\theta(\tau_i \mid h_i) = -\sum_{i=1}^N \sum_{t=1}^{T_i} \log P_\theta(\tau_i^t \mid h_i, \hat{\tau}_i^{<t}) \qquad (6)$$

To incorporate ordinal relationships into the predictions, we follow (Liu et al., 2018) to employ the Huber loss (Huber, 1992) as the regression loss $\mathcal{L}_{reg}$. The final objective is:

$$\mathcal{L}_{final} = \mathcal{L}_{NLL} + \lambda \cdot \mathcal{L}_{reg} \qquad (7)$$

where $\lambda$ is a hyperparameter that balances the two losses.

**Inference process.** The encoder processes the input $x_i$ to derive a hidden representation $h_i$ analogous to the training phase. The decoder then initiates the generation of sequence $\hat{\tau}_i$ autoregressively, beginning with the $\langle\text{SOS}\rangle$ token and proceeding until the $\langle\text{EOS}\rangle$ token is generated. Finally, the predicted value is computed by $\hat{y}_i = \sum_{t=1}^{T_i} \nu(\hat{\tau}_i^t)$.

## 3   Experiments

We first present GoR's overall performance across multiple domains, then introduce in-depth analyses of architectural choices, interval-wise performance, distributional analysis, extensibility, vocabulary ablation, and token semantics to elucidate its underlying mechanisms. Evaluation metrics include Mean Absolute Error (MAE), Cumulative Score (CS), XAUC (Zhan et al., 2022), Linear Correlation Coefficient (LCC), and Spearman's Rank Correlation Coefficient (SRCC). Due to space limit, detailed metric definitions, implementation settings, and additional results are moved to Appendix F.

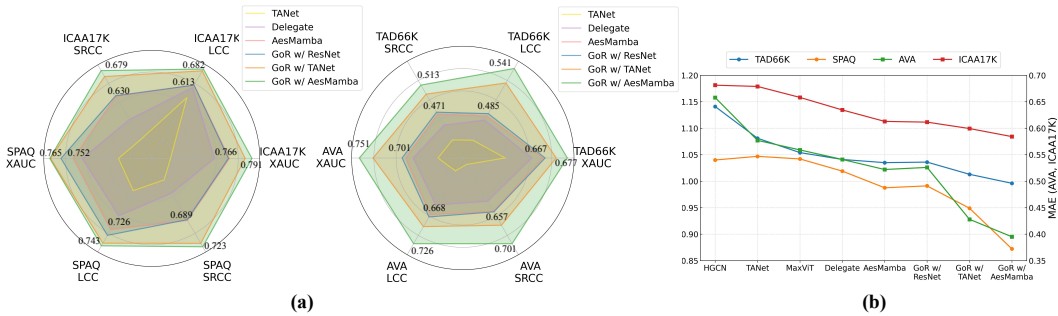

Figure 3: Aesthetics assessment results on four benchmarks (Best viewed in color).

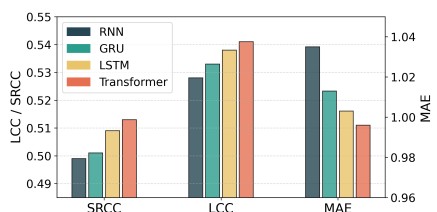

Figure 4: Performanceon the TAD66K dataset with different decoder architectures.

| Method | Criteo-SSC | | Kaggle | |
|---|---|---|---|---|
| | MAE ↓ | SRCC ↑ | MAE ↓ | SRCC ↑ |
| Two-stage (Drachen et al., 2018) | 21.719 | 0.2386 | 74.782 | 0.431 |
| MTL-MSE (Ma et al., 2018) | 21.190 | 0.2478 | 74.065 | 0.433 |
| ZILN (Wang et al., 2019) | 20.880 | 0.2434 | 72.528 | 0.524 |
| MDME (Li et al., 2022) | 16.598 | 0.2269 | 72.900 | 0.516 |
| MDAN (Liu et al., 2024) | 20.030 | 0.2470 | 73.940 | 0.437 |
| OptDist (Weng et al., 2024) | 15.784 | 0.2505 | 70.929 | 0.525 |
| HiLTV (Xu et al., 2025) | 14.764 | 0.2645 | 69.331 | 0.512 |
| **Our GoR** | **12.965** | **0.3036** | **67.075** | **0.536** |

Table 1: Performance comparison on LTV datasets with the MAE and SRCC metrics.

## 3.1 OVERALL PERFORMANCE

### 3.1.1 IMAGE AESTHETICS ASSESSMENT (IAA)

Following (He et al., 2023), we select 14 representative and state-of-the-art (SOTA) baselines for evaluation on four datasets: TAD66K (He et al., 2022), AVA (Murray et al., 2012), ICAA17K (He et al., 2023), and SPAQ (Fang et al., 2020), with four metrics: MAE, XAUC, LCC, and SRCC. Recognizing the critical importance of visual features in the IAA task (He et al., 2022), we evaluate GoR by employing three distinct encoder backbones: ResNet50 (He et al., 2016), a representative legacy architecture (He et al., 2022), and a recent SOTA model (Gao et al., 2024).

**Performance.** Fig. 3 demonstrates GoR's superiority over SOTA methods across four metrics, with three critical observations: (1) Compatibility: GoR with standard ResNet50 matches SOTA models with expert-designed architectures; (2) Robustness: Even paired with outdated TANet, GoR significant improvements over the SOTA methods—a compelling result given the critical dependence of IAA tasks on feature quality; (3) Synergy: Combined with a modern AesMamba encoder, GoR achieves new SOTA (detailed metrics in Appendix F.2.2). These results validate the universal efficacy of our generative ordinal modeling paradigm across different encoder architectures.

### 3.1.2 LIFE TIME VALUE PREDICTION (LTV)

Following (Weng et al., 2024), we evaluate GoR on the Criteo-SSC and Kaggle datasets with MAE and SRCC metrics. A Feed-Forward Network (FFN) serves as the encoder of GoR. Details regarding the encoder architecture, the baseline methods, and datasets are in Appendix F.4.

**Performance.** Across both ordinal (SRCC) and numeric (MAE) metrics, GoR consistently surpasses existing methods (see Tab. 1). It improves HiLTV by 3.25% in MAE (reduction) and 2.1% in SRCC on Kaggle. Notably, for Criteo-SSC, GoR achieves a 12.2% reduction in MAE and a 14.8% improvement in SRCC compared to the SOTA method HiLTV, substantiating the superiority of GoR.

### 3.1.3 WATCH TIME PREDICTION (WTP)

Following (Lin et al., 2023; Zhao et al., 2024), three publicly available datasets (CIKM16, KuaiRec (Gao et al., 2022a) and KuaiRand (Gao et al., 2022b)) and one industrial dataset from a real-world short-video app are used to evaluate the proposed GoR, with the metrics of MAE and

Table 2: Performance comparison among different approaches on WTP task for four datasets.

| Method | KuaiRec | | KuaiRand | | CIKM16 | | Indust. | |
|---|---|---|---|---|---|---|---|---|
| | MAE ↓ | XAUC ↑ | MAE ↓ | XAUC ↑ | MAE ↓ | XAUC ↑ | MAE ↓ | XAUC ↑ |
| VR (Value Regression) | 7.634 | 0.534 | 12.349 | 0.521 | 1.039 | 0.641 | 40.457 | 0.584 |
| WLR (Covington et al., 2016) | 6.047 | 0.545 | 11.582 | 0.529 | 0.998 | 0.672 | 36.342 | 0.588 |
| D2Q (Zhan et al., 2022) | 5.426 | 0.565 | 10.564 | 0.537 | 0.899 | 0.661 | 32.349 | 0.592 |
| CWM (Zhao et al., 2024) | 3.452 | 0.580 | 8.696 | 0.561 | 0.891 | 0.662 | 33.323 | 0.591 |
| TPM (Lin et al., 2023) | 3.456 | 0.571 | 9.573 | 0.542 | 0.850 | 0.676 | 32.437 | 0.597 |
| CREAD (Sun et al., 2024) | 3.307 | 0.594 | 9.487 | 0.549 | 0.865 | 0.678 | 29.997 | 0.604 |
| SWaT (Yang et al., 2025) | 3.438 | 0.585 | 9.553 | 0.544 | 0.857 | 0.685 | 31.245 | 0.599 |
| **Our GoR** | **3.194** | **0.616** | **7.032** | **0.567** | **0.808** | **0.696** | **28.366** | **0.611** |

Table 3: Facial age estimation results on four benchmarks

| Method | UTKFace | | FG-NET | | MORPH | | CACD | |
|---|---|---|---|---|---|---|---|---|
| | MAE ↓ | CS(%) ↑ | MAE ↓ | CS(%) ↑ | MAE ↓ | CS(%) ↑ | MAE ↓ | CS(%) ↑ |
| OR-CNN (Niu et al., 2016) | 4.40 | 63.67 | 5.09 | 83.80 | 2.83 | 61.97 | 4.01 | 73.41 |
| DLDL (Gao et al., 2017) | 4.39 | 63.65 | 5.26 | 83.83 | 2.81 | 62.43 | 3.96 | 73.37 |
| SORD (Diaz & Marathe, 2019) | 4.36 | 64.25 | 5.59 | 82.83 | 2.81 | 61.31 | 3.96 | 73.48 |
| Mean-Var. (Pan et al., 2018) | 4.42 | 63.36 | 5.45 | 83.43 | 2.83 | 62.87 | 4.07 | 72.98 |
| Unimodal (Li et al., 2022) | 4.47 | 62.67 | 5.13 | 83.97 | 2.78 | 63.15 | 4.10 | 73.55 |
| FaRL (Paplhám et al., 2024) | 3.87 | 65.38 | 4.95 | 84.52 | 3.04 | 63.49 | 3.96 | 74.18 |
| **Our GoR** | **3.43** | **66.58** | **4.68** | **85.66** | **2.69** | **64.95** | **3.73** | **75.29** |

XAUC. The encoder is consistent with the LTV task implementation in Sec. 3.1.2, and comprehensive details about datasets and compared baselines are in Appendix F.3.1.

**Performance.** We compare GoR with 6 existing methods and the results are presented in Tab. 2. Compared to the second-best method (marked in *underline*), GoR achieves relative reductions in MAE of 3.41% (KuaiRec), 19.14% (KuaiRand), and 4.94% (CIKM16), alongside relative improvements in XAUC of 3.7% (KuaiRec), 1.07% (KuaiRand), and 1.61% (CIKM16). The comprehensive improvements in both MAE and XAUC substantiate the superiority of the GoR method. Besides, GoR exhibits a 5.44% relative decrease in MAE and a 1.16% improvement in XAUC on the Indust dataset, demonstrating its potential to significantly enhance real-world user experiences.

### 3.1.4  FACIAL AGE ESTIMATION (FAE)

FAE is a discrete ordinal regression problem (e.g. 0-100 years old) unlike some previous tasks that involve continuous ordinal labels. Consistent with the datasets, baselines, and evaluation protocol used in (Paplhám et al., 2024), we evaluate GoR on four FAE datasets (UTKFace (Zhang et al., 2017), FG-NET (Lanitis et al., 2002), MORPH (Ricanek & Tesafaye, 2006), and CACD (Chen et al., 2014)) with MAE and CS (tolerance $L = 5$) metrics, and use FaRL (Zheng et al., 2022) as the encoder.

**Performance.** The results in Tab. 3 show that our GoR model achieves SOTA performance, significantly surpassing all baselines across all datasets in both evaluation metrics. GoR delivers consistent and substantial improvements — MAE reductions between 5.8% (MORPH) and 14.1% (FG-NET), and CS improvements between 1.5% (CACD) and 2.7% (FG-NET) — demonstrating its generality over diverse ordinal regression tasks, from continuous to discrete labels.

### 3.1.5  HISTORICAL IMAGE DATING (HID)

Historical Image Dating (HID) represents a typical discrete OR task, analogous to Facial Age Estimation task. Following (Wang et al., 2023), we utilize the HCI dataset (Palermo et al., 2012) with a stratified 80-5-15 split (training/validation/testing) per decade. We employ 10-fold cross-validation and report the average Mean Absolute Error (MAE) in Tab. 4. To ensure fair comparison, all methods leverage a unified ResNet50 (He et al., 2016) backbone. GoR achieves state-of-the-art performance, delivering a 3.77% MAE reduction over Ord2Seq.

## 3.2  FURTHER ANALYSIS

### 3.2.1  ARCHITECTURE-AGNOSTIC ANALYSIS

We investigate the impact of different decoder architectures on the IAA task using the TAD66K dataset, as illustrated in Fig. 4. By instantiating GoR with various sequence modeling mechanisms—including RNN (Schuster & Paliwal, 1997), GRU (Chung et al., 2014), LSTM (Hochreiter & Schmidhuber, 1997), and Transformer (Vaswani et al., 2017)—we observe a consistent trend: all

Table 4: Performance comparison among different methods on the HCI Dataset.

| Methods Datasets | Palermo et al. (Palermo et al., 2012) | CNNPOR (Liu et al., 2018) | GP-DNNOR (Liu et al., 2019) | SORD (Diaz & Marathe, 2019) | POE (Li et al., 2021) | MWR (Shin et al., 2022) | Ord2Seq (Wang et al., 2023) | GoR |
|---|---|---|---|---|---|---|---|---|
| HCI  MAE ↓ | 0.93 | 0.82 | 0.76 | 0.70 | 0.66 | 0.58 | 0.53 | **0.51** |

Table 5: MAE comparison across varying Ground Truth (GT) time intervals on the KuaiRec dataset.

| Method | Time Intervals (Seconds) | | | | | |
|---|---|---|---|---|---|---|
| | 0-2s | 2-4s | 4-6s | 6-8s | 8-10s | >10s |
| TPM | 7.70 | 5.92 | 3.79 | 1.72 | 1.37 | **6.25** |
| CREAD | 7.07 | 4.96 | 3.14 | 1.28 | 1.26 | 6.71 |
| GoR | **5.50** | **3.95** | **2.28** | **1.02** | **1.06** | 6.26 |

Table 6: Statistical comparison (Mean and Var) of the predicted distributions.

| Distribution | Mean ($\mu$) | Var ($\sigma^2$) |
|---|---|---|
| GT | 7.69 | 11.46 |
| TPM | 9.11 | 2.56 |
| CREAD | 8.54 | 1.81 |
| GoR | **7.73** | 2.88 |

variants successfully outperform current SOTA methods. While the Transformer-based variant yields the optimal performance, the universal improvement across all architectures strongly validates the robustness and architectural independence of our GoR framework.

### 3.2.2 PREDICTIONS ANALYSIS

To evaluate fine-grained predictive capabilities, we dissect model performance across varying watch-time segments on the KuaiRec dataset, where approximately 80% of instances possess a Ground Truth (GT) of under 10 seconds. As detailed in Tab. 5, GoR demonstrates substantial superiority over CREAD and TPM across all high-frequency short-to-medium duration brackets (0-10s). Although TPM exhibits a marginal edge in the sparse >10s regime, GoR consistently minimizes the Mean Absolute Error (MAE) for the vast majority of user interactions.

We further scrutinize the global statistical properties of the predictions in Tab. 6. GoR achieves exceptional calibration, generating a predictive distribution whose mean ($\mu = 7.73$) aligns remarkably well with the GT ($\mu = 7.69$). This statistical fidelity is largely attributed to GoR's coarse-to-fine tokenization, which dynamically adapts to varying granularities. Conversely, both CREAD and TPM suffer from pronounced overestimation biases, outputting significantly higher means (8.54 and 9.11, respectively). This systemic overestimation inherently stems from their reliance on rigid discretization boundaries, where broad intervals in tail buckets disproportionately inflate errors for the more prevalent, shorter GTs.

### 3.2.3 VOCABULARY ANALYSIS

We investigate the impact of vocabulary initialization by comparing our proposed approach against two baseline strategies: a manual strategy (defined by intervals of $[1, 3, 5, 7] \times 10^n$) and a binary strategy (using power-of-two intervals $2^n$ starting from 1). As detailed in Tab. 7, our analysis reveals three key findings: (1) The quantile-based method consistently outperforms both manual and binary strategies, primarily benefiting from a more balanced token distribution (see Fig. 5(a)). (2) The integration of CoDi universally enhances all initialization strategies. This improvement is attributed to CoDi's ability to yield a more uniform token frequency distribution, thereby mitigating token-level variance and per-step bias $B$, which perfectly aligns with our theoretical expectations. (3) The $\beta$ sensitivity analysis in Fig. 5(b) demonstrates a non-monotonic performance trend as $\beta$ decreases and more tokens are filtered. This phenomenon exemplifies the classical bias-variance trade-off: an initial reduction in vocabulary size effectively suppresses variance, whereas excessive compression drastically increases bias, strictly corroborating Theorem 1.

### 3.2.4 COMPATIBILITY WITH GENERATIVE OPTIMIZATION STRATEGIES.

As discussed in Sec. 1, GoR can seamlessly accommodate optimization strategies for generative language model with negligible adaptation cost. Here, we assess this compatibility by incorporating several representative strategies prevalent in language models: 1) Teacher Forcing (TF) (Sutskever, 2014) is a training method that feeds the ground-truth token at step $t - 1$ into the decoder at step $t$. 2) Curriculum Learning (CL) (Bengio et al., 2015) progressively shifts the training strategy from full TF to a strategy that closely mimics autoregressive inference. 3) N-gram Prediction (NP) (Goodman et al., 2020) simultaneously predicts N tokens to improve predictive lookahead and compensate for

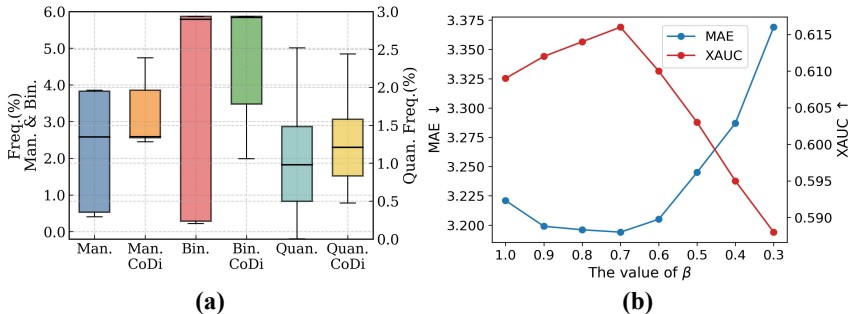

Figure 5: Vocabulary analysis on KuaiRec dataset for WTP task: (a) Token frequency distributions under different vocabulary strategies. (b) The sensitivity analysis of $\beta$.

Table 7: Performance Across Vocabulary Construction Strategies (w/ vs. w/o CoDi)

| Vocabulary design | KuaiRec | | CIKM16 | |
|---|---|---|---|---|
| | MAE ↓ | XAUC ↑ | MAE ↓ | XAUC ↑ |
| Manual (Man.) | 3.281 | 0.604 | 0.825 | 0.685 |
| Binary (Bin.) | 3.268 | 0.605 | 0.821 | 0.687 |
| Quantile (Quan.) | 3.221 | 0.609 | 0.820 | 0.688 |
| Man.+CoDi | 3.253 | 0.610 | 0.819 | 0.689 |
| Bin.+CoDi | 3.239 | 0.611 | 0.815 | 0.691 |
| Quan.+CoDi | 3.194 | 0.616 | 0.808 | 0.696 |

Table 8: Compatibility with existing generative optimization strategies on WTP and LTV tasks.

| | Strategy | | | KuaiRec | | Criteo-SSC | |
|---|---|---|---|---|---|---|---|
| | TF | CL | NP | MAE ↓ | XAUC ↑ | MAE ↓ | SRCC ↑ |
| (a) | ✓ | | | 3.359 | 0.588 | 16.198 | 0.252 |
| (b) | | ✓ | | 3.299 | 0.592 | 14.893 | 0.264 |
| (c) | | | ✓ | 3.241 | 0.604 | 13.996 | 0.276 |
| (d) | ✓ | | ✓ | 3.208 | 0.612 | 13.068 | 0.292 |
| (e) | | ✓ | ✓ | 3.194 | 0.616 | 12.965 | 0.304 |
| (e) w/ DPO | - | - | - | 3.185 | 0.620 | 12.438 | 0.309 |

output-to-input gradients. 4) DPO[3] (Rafailov et al., 2023) is a reinforcement-learning-based strategy that optimizes model outputs via preference alignment. Our GoR requires no explicit reward model, instead using beam-search candidates with MAE against labels as implicit preference signals. The results in Tab. 8 indicate that this compatibility significantly enhances model performance. Crucially, these improvements are achieved without introducing additional model parameters, demonstrating a cost-effective and scalable enhancement that facilitates GoR's future exploration in wider applications.

# 4 CONCLUSION

In this paper, we propose **GoR**, the first generative framework tailored for Ordinal Regression (OR). Diverging from traditional rigid binning methods, GoR innovatively formulates OR as an autoregressive sequence generation task. By predicting tokens for ordinal value segments in a step-by-step manner, this paradigm explicitly models inherent sequential dependencies and enables adaptive resolution. Our methodology is rigorously supported by a comprehensive bias-variance theoretical analysis, which motivates the proposed CoDi metric for optimal vocabulary construction. Empirically, GoR consistently achieves state-of-the-art (SOTA) performance across 15 diverse benchmarks spanning 5 distinct domains, highlighting its remarkable robustness and generalizability. Ultimately, GoR not only unlocks the potential of generative architectures for ordinal data but also establishes a strong, versatile baseline for future generative OR research.

## 4.1 LIMITATION AND FUTURE

While GoR establishes a novel generative paradigm for ordinal regression and achieves state-of-the-art performance, several limitations exist, opening promising avenues for future research. First, the autoregressive nature of GoR, while enabling sequential modeling, incurs an inference latency cost that is proportional to the output sequence length. This poses a challenge for tasks requiring rapid prediction of long sequences. Second, GoR, like other language generation models, is susceptible to the risk of error accumulation. Errors in predicting earlier tokens can compound in subsequent steps, potentially leading to larger deviations in the final prediction. Exploring sequence-level optimization or calibration strategies, such as those based on reinforcement learning, could help alleviate this issue.

---

[3]To avoid the prolonged training time associated with reinforcement-learning-based post-training, while effective, all results are reported using the efficient setup from line (e).

## ACKNOWLEDGMENTS

This work was partially supported by Kuaishou Technology. The computations in this research were partially performed using the CFFF platform of Fudan University.

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

APPENDIX

## A  THE USE OF LARGE LANGUAGE MODELS (LLMS)

Generative AI tools (e.g., ChatGPT) were used solely to improve the manuscript's clarity and readability during the writing stage. These tools were not employed for generating any novel content, such as text, figures, tables, code, or experimental results. No generative AI was used in the conception, implementation, analysis, or evaluation of the research itself. The authors take full responsibility for the integrity and accuracy of the final manuscript.

## B  RELATED WORK

### B.1  ORDINAL REGRESSION (OR)

OR addresses prediction tasks with ordered targets, widely applied in diverse domains like facial age estimation (Niu et al., 2016; Chen et al., 2017; Feng et al., 2026), image aesthetic/quality assessment (He et al., 2022; 2023), watch-time prediction Ma et al. (2024); Sun et al. (2024); Lin et al. (2023); Jin et al. (2026), life-time value prediction (Wang et al., 2019; Li et al., 2022; Weng et al., 2024). Prior OR works have predominantly relied on Continuous Space Discretization (CSD) (Wang et al., 2025), transforming OR into classification. Within the CSD paradigm, key directions include methods that enhance boundary discrimination via reference comparisons (Shin et al., 2022; Li et al., 2021; Zheng et al., 2024), but are sensitive to heuristic reference selection. Another prevalent rank-based approach implicitly encodes ordinality via label transformation into sequential binary subtasks (Niu et al., 2016; Lin et al., 2023; Sun et al., 2024; Chen et al., 2017; Liu et al., 2018). While effective, the fixed discretization leads to prediction rigidity and can amplify errors for head categories, particularly in long-tailed distributions. Recent CLIP-based works align image features with textual ordinal descriptions to enhance the semantic understanding and generalization of ordinal relationships, representing a rapidly growing and promising research direction. The Learnable Prompts strategy (Li et al., 2022; Yu et al., 2024) employs trainable context vectors to automatically capture ordinal relationships and extract rank concepts from CLIP's latent space. Conversely, Semantic Alignment approaches (Wang et al., 2023; Du et al., 2024) focus on constructing rank-specific textual descriptions. In contrast, GoR adopts a fundamentally different, generative perspective of autoregressive sequence modeling in language models, which inherently captures explicit sequential dependencies and employs a dynamic termination mechanism to generate sequences of variable lengths. This grants significantly greater flexibility in output granularity compared to methods constrained by fixed bins, enabling adaptation to varying data distributions and prediction requirements.

### B.2  SEQUENCE PREDICTION

Sequence prediction, which necessitates models to comprehend input context and produce output sequences, initially focuses on natural language processing (NLP) tasks such as machine translation (Sutskever, 2014; Cho, 2014) and text summarization (Wang et al., 2019; Xiao et al., 2020; Liu et al., 2025). With the advancement of deep learning (Hui et al., 2026; Gu et al., 2026; Ma et al., 2025; Liu et al., 2025; 2024a;b;c; LIU et al., 2026), the advent of the Transformer architecture (Vaswani et al., 2017; Ma et al., 2025; Zhang et al., 2025; Kang et al., 2025; Liu et al., 2023a;b; Zhang et al., 2026) significantly improves sequence prediction capabilities, leading to numerous derivative models (Liu et al., 2019; Yang et al., 2019; Zhou et al., 2025) and expanding its application to fields such as computer vision (CV) (Tian et al., 2024; Wang et al., 2024; Zhou et al., 2026) and recommendation systems (Hidasi, 2015; Sun et al., 2019) through the successful reformulation of their tasks as effective end-to-end sequence prediction problems. However, sequence prediction has not yet been applied to OR, and our work pioneers a sequence prediction perspective for OR, offering a fundamentally novel modeling paradigm. A related work is Ord2Seq (Wang et al., 2023). It maps generated sequences to fixed bins and, in essence, remains sequential binary subtasks under the rank-based paradigm and does not scale well when the number of categories or the value range is large—its reported validation has been limited to at most eight ordinal groups. In contrast, GoR employs a generative autoregressive formulation with a dynamic $\langle \text{EOS} \rangle$, enabling adaptive ordinal segmentation rather than relying on predefined bins.

### B.3 TOKENIZER DESIGN

Tokenizer design is widely employed in generative language models for compact vocabulary representation and broadly falls into two categories: bottom-up merging and top-down pruning. The former, exemplified by BPE (Sennrich et al., 2016) and WordPiece (Wu et al., 2016), iteratively combines subword units based on statistical criteria. Conversely, top-down pruning methods, represented by Unigram (Kudo & Richardson, 2018), reduce large initial sets by evaluating and eliminating subwords according to their contributions. Building upon these foundational strategies, recent literature has explored various enhancements and adaptations (Xu et al., 2021; Hofmann et al., 2022; Yehezkel & Pinter, 2023; Schmidt et al., 2024), further improving tokenization efficiency. However, these traditional NLP tokenization strategies are not directly applicable to our GoR, where tokens inherently exhibit dual sequential and numerical additive semantics, necessitating customized methodologies. While a related work (Ma et al., 2024) attempts to address similar issues, it remains a domain-specific method rather than a general-purpose framework. Furthermore, it relies on a heuristic vocabulary design without rigorous theoretical optimization, which inherently limits its modeling accuracy.

## C PROOF OF PROPOSITION 1

### C.1 THEORETICAL ANALYSIS

This section provides a theoretical analysis of these limitations, demonstrating the importance of capturing temporal dependencies to improve prediction accuracy.

Let $\{(x_i, y_i)\}_{i=1}^N$ be the training set. Discretize the value range of $y_i$ into $M$ intervals $d_m = [c_{m-1}, c_m]$ with boundaries $c_0 < \cdots < c_M$. Define binary variables $\mathbf{B}_i^m = 1(y_i > c_m)$ for $m = 1, \ldots, M$, and let $\mathbf{B}_i = \{\mathbf{B}_i^1, \ldots, \mathbf{B}_i^M\}$ with decision history $\mathbf{B}_i^{<m} = (\mathbf{B}_i^1, \ldots, \mathbf{B}_i^{m-1})$.

The label transformation methods with sequential binary subtasks implicitly assume conditional independence across these discretized intervals:

$$P_{\text{naive}}(\mathbf{B}_i \mid x_i) = \prod_{m=1}^M P(\mathbf{B}_i^m \mid x_i). \tag{8}$$

In contrast, the true conditional distribution factorizes sequentially:

$$P_{\text{true}}(\mathbf{B}_i \mid x_i) = \prod_{m=1}^M P(\mathbf{B}_i^m \mid \mathbf{B}_i^{<m}, x_i). \tag{9}$$

The KL divergence between these two distributions is given by:

$$
\begin{aligned}
D_{KL}(P_{\text{true}} \| P_{\text{naive}}) &= \sum_{\mathbf{B}_i} P_{\text{true}}(\mathbf{B}_i \mid x_i) \log \frac{P_{\text{true}}(\mathbf{B}_i \mid x_i)}{P_{\text{naive}}(\mathbf{B}_i \mid x_i)} \\
&= \sum_{\mathbf{B}_i} P_{\text{true}}(\mathbf{B}_i \mid x_i) \log \frac{\prod_{m=1}^M P(\mathbf{B}_i^m \mid \mathbf{B}_i^{<m}, x_i)}{\prod_{m=1}^M P(\mathbf{B}_i^m \mid x_i)} \\
&= \sum_{\mathbf{B}_i} P_{\text{true}}(\mathbf{B}_i \mid x_i) \sum_{m=1}^M \log \frac{P(\mathbf{B}_i^m \mid \mathbf{B}_i^{<m}, x_i)}{P(\mathbf{B}_i^m \mid x_i)}.
\end{aligned}
\tag{10}
$$

Rearranging the summation terms, we derive the total KL divergence that quantifies the error introduced by ignoring dependencies among discretized intervals:

$$
\begin{aligned}
D_{KL}(P_{\text{true}} \| P_{\text{naive}}) &= \sum_{m=1}^M \sum_{\mathbf{B}_i} P_{\text{true}}(\mathbf{B}_i \mid x_i) \log \frac{P(\mathbf{B}_i^m \mid \mathbf{B}_i^{<m}, x_i)}{P(\mathbf{B}_i^m \mid x_i)} \\
&= \sum_{m=1}^M \mathbb{E}_{\mathbf{B}_i \sim P_{\text{true}}} \left[ \log \frac{P(\mathbf{B}_i^m \mid \mathbf{B}_i^{<m}, x_i)}{P(\mathbf{B}_i^m \mid x_i)} \right].
\end{aligned}
\tag{11}
$$

We can decompose this expectation using the Law of Iterated Expectations:

$$D_{KL}(P_{\text{true}} \| P_{\text{naive}}) = \sum_{m=1}^{M} \mathbb{E}_{\mathbf{B}_i^{<m}} \left[ \mathbb{E}_{\mathbf{B}_i^{\geq m} | \mathbf{B}_i^{<m}} \left[ \log \frac{P(\mathbf{B}_i^m | \mathbf{B}_i^{<m}, \mathbf{x}_i)}{P(\mathbf{B}_i^m | \mathbf{x}_i)} \right] \right]$$

$$= \sum_{m=1}^{M} \mathbb{E}_{\mathbf{B}_i^{<m}} \left[ \sum_{b^m \in \{0,1\}} P(\mathbf{B}_i^m = b^m | \mathbf{B}_i^{<m}, \mathbf{x}_i) \log \frac{P(\mathbf{B}_i^m = b^m | \mathbf{B}_i^{<m}, \mathbf{x}_i)}{P(\mathbf{B}_i^m = b^m | \mathbf{x}_i)} \right] \quad (12)$$

$$= \sum_{m=1}^{M} \mathbb{E}_{\mathbf{B}_i^{<m}} \left[ D_{KL}^{(m)} \right].$$

This can be explicitly expressed as the conditional KL divergence for each bucket:

$$D_{KL}^{(m)} = D_{KL} \left( P(\mathbf{B}_i^m \mid x_i, \mathbf{B}_i^{<m}) \,\|\, P(\mathbf{B}_i^m \mid x_i) \right)$$

$$= \sum_{b^m \in \{0,1\}} P(\mathbf{B}_i^m = b^m \mid x_i, \mathbf{B}_i^{<m}) \log \frac{P(\mathbf{B}_i^m = b^m \mid x_i, \mathbf{B}_i^{<m})}{P(\mathbf{B}_i^m = b^m \mid x_i)}. \quad (13)$$

This expectation can be expressed as:

$$\mathbb{E}_{\mathbf{B}_i^{<m}} \left[ D_{KL}^{(m)} \right] = I(\mathbf{B}_i^m; \mathbf{B}_i^{<m} \mid x_i), \quad (14)$$

where $I(\mathbf{B}_i^m; \mathbf{B}_i^{<m} \mid x_i)$ denotes the conditional mutual information between the current decision $\mathbf{B}_i^m$ and all preceding decisions $\mathbf{B}_i^{<m}$, given the features $x_i$. The detailed proof is in the following:

$$I(\mathbf{B}_i^m; \mathbf{B}_i^{<m} \mid x_i) \triangleq \sum_{\mathbf{b}^{<m}, b^m} P(\mathbf{b}^{<m}, b^m \mid x_i) \log \frac{P(b^m \mid \mathbf{b}^{<m}, x_i)}{P(b^m \mid x_i)}$$

$$= \sum_{\mathbf{b}^{<m}} P(\mathbf{b}^{<m} \mid x_i) \sum_{b^m} P(b^m \mid \mathbf{b}^{<m}, x_i) \log \frac{P(b^m \mid \mathbf{b}^{<m}, x_i)}{P(b^m \mid x_i)} \quad (15)$$

$$= \sum_{\mathbf{b}^{<m}} P(\mathbf{b}^{<m} \mid x_i) \left( D_{KL} \left( P(\mathbf{B}_i^m \mid x_i, \mathbf{B}_i^{<m}) \,\|\, P(\mathbf{B}_i^m \mid x_i) \right) \right)$$

$$\triangleq \mathbb{E}_{\mathbf{B}_i^{<m}} \left[ D_{KL}^{(m)} \right].$$

The derived KL divergence decomposition illustrates that the error introduced by the naive discretized modeling approach which ignores dependencies across intervals, can be quantified precisely as the cumulative sum of conditional mutual information across all discretized intervals. Specifically, if intervals are entirely independent (i.e., mutual information $I = 0$), the resulting KL divergence error is zero; conversely, if strong dependencies exist among intervals ($I > 0$), the error increases proportionally to the strength of these dependencies.

## C.2 EMPIRICAL VALIDATION

To empirically validate Proposition 1, we conduct additional experiments using an SOTA baseline SWaT (Yang et al., 2025) on the CIKM16 dataset of watch-time prediction task. Specifically, we introduce explicit sequential dependencies by modeling previous bin features with an RNN for joint prediction. The results in Tab. 9 demonstrate performance gains, substantiating our theoretical claims.

Table 9: Empirical validation of Proposition 1 on the CIKM16 dataset.

| Method | MAE ↓ | XAUC ↑ |
|---|---|---|
| SWaT | 0.857 | 0.685 |
| SWaT+RNN | 0.831 | 0.689 |

# D PROOF OF THEOREM 1

We provide a formal bias-variance decomposition of the expected squared error in GoR. Our goal is to upper bound the prediction error of a model that generates a sequence of value tokens.

Let the true label be defined as:

$$y_i = \sum_{t=1}^{T_i} \nu(\tau_i^t), \tag{16}$$

and the predicted label be:

$$\hat{y}_i = \sum_{t=1}^{T_i} \nu(\hat{\tau}_i^t), \tag{17}$$

where $T_i$ represents the overall length of the token sequence for sample $i$, $\tau_i^t$ and $\hat{\tau}_i^t$ denote the ground-truth and predicted tokens at step $t$, and $\nu(\cdot)$ maps a token to its corresponding numeric value.

We model $\nu(\tau_i^t)$ as a discrete random variable, denoted as $C_i^t$. The probability distribution of $C_i^t$ is given by: $P(C_i^t = \omega), \omega \in \{\nu(\omega_j)\}_{j=1}^V$, which denotes the probability of $C_i^t$ taking the value $\omega$. So, the predicted token output be modeled as $\hat{y}_i = \sum_{t=1}^{T_i} \hat{C}_i^t$

We now analyze the expected squared error:

$$\mathbb{E}[(\hat{y}_i - y_i)^2] = \mathbb{E}\left[\left(\sum_{t=1}^{T_i} \hat{C}_i^t - \sum_{t=1}^{T_i} C_i^t\right)^2\right]. \tag{18}$$

Let $\Delta_t := \hat{C}_i^t - C_i^t$. Then we can write:

$$\mathbb{E}\left[\left(\sum_{t=1}^{T_i} \Delta_t\right)^2\right] = \underbrace{\left(\sum_{t=1}^{T_i} \mathbb{E}[\Delta_t]\right)^2}_{\text{Bias}^2} + \underbrace{\mathbb{V}\left(\sum_{t=1}^{T_i} \Delta_t\right)}_{\text{Variance}} \tag{19}$$

$$= \left(\sum_{t=1}^{T_i} b_t\right)^2 + \sum_{t=1}^{T_i} \mathbb{V}(\Delta_t) + \sum_{t \neq t'} \text{Cov}(\Delta_t, \Delta_{t'}), \tag{20}$$

where $b_t := \mathbb{E}[\Delta_t] = \mathbb{E}[\hat{C}_i^t] - C_i^t$. This formulation captures the compounding errors of autoregressive models through the covariance term, which reflects correlations between errors at different steps. Next, we analyze the bias and variance terms separately to derive their upper bounds.

**Bias Upper Bound.** If the model is unbiased at each step, $b_t = 0$. Otherwise, we assume:

$$|b_t| \leq B, \quad \forall t, \tag{21}$$

where $B$ represents the maximum bias across all time steps in the prediction sequence, mainly governed by the model's predictive accuracy. In practice, $B$ corresponds to the extreme case where the predicted token deviates most from the ground-truth token at the current step. Hence, it can be bounded by the largest token value at the current step $B \leq \max_{1 \leq t \leq T_i} C_i^t$. Then,

$$\left(\sum_{t=1}^{T_i} b_t\right)^2 \leq T_i^2 B^2. \tag{22}$$

**Variance Upper Bound.** Applying the Cauchy–Schwarz inequality:

$$\sum_{t \neq t'} \text{Cov}(\Delta_t, \Delta_{t'}) \leq \sum_{t \neq t'} \sqrt{\mathbb{V}(\Delta_t)\mathbb{V}(\Delta_{t'})} \leq \frac{T_i(T_i - 1)}{2} \cdot \max_t \mathbb{V}(\Delta_t). \tag{23}$$

We then analyze the item $\max_t \mathbb{V}(\Delta_t)$.

$$\mathbb{V}(\Delta_t) = \mathbb{V}\left(\hat{C}_i^t - C_i^t\right) = \mathbb{V}\left(\hat{C}_i^t\right) + \mathbb{V}\left(C_i^t\right) - 2\text{Cov}\left(\hat{C}_i^t, C_i^t\right) \tag{24}$$

Assume that the predicted variable and the true variable are two independent random variables. Since the vocabulary is the same, the range of values for both the predicted and the true items is identical. Assuming token values are bounded in $[\omega_{\min}, \omega_{\max}]$, we apply Popoviciu's inequality:

$$\mathbb{V}(\hat{C}_i^t) \leq \frac{(\omega_{\max} - \omega_{\min})^2}{4}. \tag{25}$$

Let:

$$V_{var} := \max_t \mathbb{V}(\Delta_t) \leq \frac{(\omega_{\max} - \omega_{\min})^2}{4}. \tag{26}$$

Then total variance becomes:

$$\mathbb{V}(\sum_t \Delta_t) \leq T_i^2 V_{var}. \tag{27}$$

**Final Bound.** Combining both components, we obtain:

$$\mathbb{E}\left[(\hat{y}_i - y_i)^2\right] \leq T_i^2 B^2 + T_i^2 \cdot \frac{(\omega_{\max} - \omega_{\min})^2}{4} \tag{28}$$

This theoretical bound reveals three critical insights for GoR optimization: (1) prediction error grows quadratically with sequence length $T_i$, suggesting shorter sequences are preferable when possible; (2) both bias and variance contribute proportionally to overall error, necessitating balanced optimization; and (3) token value range $(\omega_{\max} - \omega_{\min})$ directly impacts variance, indicating that carefully designed vocabularies with appropriate value distributions can substantially improve model performance. These findings provide a principled foundation for our vocabulary construction strategy.

## E  QUANTILE-BASED VOCABULARY INITIALIZATION STRATEGY

This section details the Quantile-based Vocabulary Initialization Strategy adopted in Sec. 2.4.1. As shown in Alg. 2, this iterative strategy constructs the vocabulary by selecting tokens based on a fixed percentile $q$ of the remaining label values, subtracting them from exceeding values, and repeating until residuals are negligible. Alg. 2 serves as the initialization stage: it provides a coarse yet comprehensive token set. Building on this, Alg. 1 in Sec. 2.4.1 further prunes and refines the vocabulary via the CoDi criterion, yielding a compact and task-adaptive representation.

## F  ADDITIONAL EXPERIMENTS

### F.1  EXPERIMENTAL SETTINGS

#### F.1.1  METRICS.

A set of performance metrics is utilized to evaluate the proposed method across various tasks. Task requirements determine the specific metrics applied for each task:

- **MAE (Mean Absolute Error)**: This regression precision is measured as the average absolute error between the value prediction $\{\hat{y}_i\}_{i=1}^N$ and the ground truth $\{y_i\}_{i=1}^N$ and is formulated as $\frac{1}{N}\sum_{i=1}^N |y_i - \hat{y}_i|$.
- **CS (Cumulative Score)**: This metric quantifies the proportion of instances in the test set for which the absolute error between the predicted value $\hat{y}_i$ and the ground truth value $y_i$ is less than or equal to a specified tolerance $L$.
- **XAUC** (Zhan et al., 2022): This metric measures the agreement between the predicted ranks and the ground truth order for pairs of samples. Calculated over uniformly sampled pairs, XAUC represents the proportion of pairs where the predicted relative order is consistent with the true relative order. Higher XAUC indicates superior performance in capturing ordinal relationships.
- **LCC (Linear Correlation Coefficient)** (Talebi & Milanfar, 2018): This metric quantifies the linear relationship between the predicted values $\{\hat{y}_i\}_{i=1}^N$ and the ground truth values $\{y_i\}_{i=1}^N$. It is computed as their covariance divided by the product of their standard deviations. The LCC ranges in $[-1, 1]$, with values closer to $\pm 1$ indicating a stronger linear correlation.

---

**Algorithm 2:** Quantile-based Vocabulary Initialization

---

**Input:** Dataset labels $Y = \{y_i\}_{i=1}^N$; Precision threshold $\varepsilon$; Fixed percentile $q$
**Output:** Initial vocabulary $\mathcal{W}$

```
// Initialization
```
1 Sort $Y$ in descending order to obtain residual set $\hat{Y} = \{\hat{y}_1, \ldots, \hat{y}_N\}$
2 $\mathcal{W} \leftarrow \varnothing, k \leftarrow 1, err \leftarrow \infty$
3 **while** $err > \varepsilon$ **do**
4     Compute the $q$-percentile $z_k$ of $\hat{Y}$
5     **if** $z_k = 0$ **then**
6         **break**             // Terminate if no positive gain available
7     **end**
8     $\mathcal{W} \leftarrow \mathcal{W} \cup \{z_k\}$
```
   // Update residual labels
```
9     **foreach** $\hat{y}_i \in \hat{Y}$ **do**
10         **if** $\hat{y}_i \geq z_k$ **then**
11             $\hat{y}_i \leftarrow \hat{y}_i - z_k$
12         **end**
13     **end**
```
   // Evaluate relative convergence
```
14     $err \leftarrow \max_i \frac{\hat{y}_i}{y_i}$
15     $k \leftarrow k + 1$
16 **end**
17 **return** $\mathcal{W}$

---

- **SRCC (Spearman's Rank Correlation Coefficient) (Talebi & Milanfar, 2018)**: SRCC assesses the monotonic relationship between the ranks of the predicted values $\{\hat{y}_i\}_{i=1}^N$ and the ranks of the ground truth values $\{y_i\}_{i=1}^N$. As a non-parametric measure of rank correlation, it ranges in $[-1, 1]$. Values closer to $\pm 1$ indicate a stronger monotonic correlation. SRCC is less sensitive to outliers compared to LCC.

### F.1.2 IMPLEMENTATION DETAILS.

Unless otherwise specified in the respective experimental sections, the following training protocol is adopted. The proposed GoR architecture employs an encoder-decoder framework. The encoder architecture in GoR is tailored to the specific task, with details regarding data processing and encoder configurations provided in the corresponding experimental sections. The decoder in GoR is a two-layer Transformer decoder utilizing a 4-head attention mechanism. The hyperparameter $\lambda$ in Eq. (7) is set to 10. For vocabulary construction, in Alg. 2 for initial vocabulary, $q$ is set to 0.9, $\varepsilon$ is set to 0.005. In Alg. 1, $\beta$ is set to 0.7 and $\epsilon$ is set to 0.005. To mitigate overfitting, a dropout rate of 0.1 is applied. The Adam optimizer (Kingma & Ba, 2014) with default parameters ($\beta_1 = 0.9$, $\beta_2 = 0.999$) and a learning rate of 5e-4 are used to minimize the objective function. For experiments involving image data (common in computer vision tasks), training is conducted for 100 epochs with a batch size of 128. For the tasks involving structured data (WTP and LTV), training is performed for 20 epochs using a batch size of 1024. Experiments are conducted on a system equipped with an NVIDIA RTX 4090 GPU.

### F.1.3 REPRODUCIBILITY AND FAIR COMPARISON.

Each experiment is repeated five times, and we report both the mean and standard deviation (See Appendix. F.5). As noted in prior work (Paplhám et al., 2024), state-of-the-art methods in the FAE field often exhibit large performance variance due to inconsistent dataset splits, preprocessing protocols, and evaluation criteria, rendering many results incomparable and irreproducible. Motivated by this, (Paplhám et al., 2024) proposed a standardized evaluation protocol. We observe the same issue in the IAA domain Consequently, we reproduced all baselines under their original hyperparameter settings in their respective papers and averaged the results. This strategy offers two key benefits: (1) it

Table 10: The results of Image Aesthetics Assessment task on TAD66K and AVA datasets

| Method | TAD66K | | | | AVA | | | |
|---|---|---|---|---|---|---|---|---|
| | MAE ↓ | XAUC ↑ | LCC ↑ | SRCC ↑ | MAE ↓ | XAUC ↑ | LCC ↑ | SRCC ↑ |
| RAPID (Lu et al., 2014) | 1.766 | 0.510 | 0.332 | 0.314 | 0.978 | 0.513 | 0.336 | 0.327 |
| AADB (Kong et al., 2016) | 1.463 | 0.523 | 0.400 | 0.379 | 0.784 | 0.534 | 0.431 | 0.408 |
| PAM (Ren et al., 2017) | 1.314 | 0.534 | 0.440 | 0.422 | 0.614 | 0.619 | 0.531 | 0.521 |
| NIMA (Talebi & Milanfar, 2018) | 1.422 | 0.511 | 0.405 | 0.390 | 0.715 | 0.532 | 0.472 | 0.447 |
| ALamp (Ma et al., 2017) | 1.349 | 0.523 | 0.422 | 0.411 | 0.657 | 0.579 | 0.498 | 0.487 |
| $MP_{ada}$ (Sheng et al., 2018) | 1.191 | 0.589 | 0.408 | 0.389 | 0.602 | 0.632 | 0.543 | 0.531 |
| MLSP (Hosu et al., 2019) | 1.132 | 0.620 | 0.432 | 0.409 | 0.579 | 0.657 | 0.563 | 0.553 |
| BIAA (Zhu et al., 2020) | 1.329 | 0.538 | 0.431 | 0.348 | 0.672 | 0.566 | 0.496 | 0.476 |
| UIAA (Zeng et al., 2019) | 1.281 | 0.548 | 0.441 | 0.361 | 0.608 | 0.626 | 0.535 | 0.525 |
| HGCN (She et al., 2021) | 1.141 | 0.615 | 0.419 | 0.406 | 0.658 | 0.578 | 0.511 | 0.486 |
| TANet (He et al., 2022) | 1.081 | 0.649 | 0.452 | 0.428 | 0.577 | 0.659 | 0.568 | 0.554 |
| MaxViT (Tu et al., 2022) | 1.054 | 0.659 | 0.472 | 0.441 | 0.559 | 0.679 | 0.594 | 0.571 |
| Delegate (He et al., 2023) | 1.041 | 0.661 | 0.477 | 0.451 | 0.541 | 0.688 | 0.642 | 0.634 |
| AesMamba (Gao et al., 2024) | 1.035 | 0.666 | 0.482 | 0.468 | 0.522 | 0.697 | 0.663 | 0.656 |
| **GoR with ResNet** | 1.036 | 0.667 | 0.485 | 0.471 | 0.526 | 0.701 | 0.668 | 0.657 |
| **GoR with TANet** | 1.013 | 0.672 | 0.523 | 0.499 | 0.428 | 0.735 | 0.689 | 0.686 |
| **GoR with AesMamba** | 0.996 | 0.677 | 0.541 | 0.513 | 0.395 | 0.751 | 0.726 | 0.701 |

ensures fair and consistent comparisons; and (2) it establishes GoR as a reliable baseline to facilitate standardized evaluations in future research.

## F.2 Image Aesthetics Assessment (IAA)

### F.2.1 Datasets, Baselines, and Experimental Setup.

GoR is evaluated on four widely used IAA datasets: TAD66K (He et al., 2022), AVA (Murray et al., 2012), ICAA17K (He et al., 2023), and SPAQ (Fang et al., 2020). Data was randomly split into 80% for training, 10% for validation, and 10% for testing. Due to the relatively small range of aesthetics scores (typically 0-10), labels were scaled by 100 for GoR's vocabulary construction and ordinal target sequencing. Predictions were scaled back by 100 for evaluation metric computation to ensure fair comparison.

Baselines were chosen based on two criteria: 1) classical architectures with available code, and 2) state-of-the-art (SOTA) performance in specific areas, such as personalized IAA. For a fair comparison, these baselines were trained using their recommended hyperparameter settings and evaluated under identical training and testing configurations. Consistent with the approach in (He et al., 2023), all compared baselines were subjected to identical data preprocessing.

Given the critical role of visual features in image aesthetics assessment, we evaluate GoR by employing three different encoder backbones: ResNet50 (He et al., 2016) as a widely recognized standard, a representative older architecture (He et al., 2022), and a recent SOTA model (Gao et al., 2024). This strategy also helps to ensure that observed performance gains are due to the GoR framework itself, rather than simply an increase in model parameters.

### F.2.2 Comprehensive Baselines Comparison for Image Aesthetics Assessment.

Due to space constraints in Sec. 3.1.1 of the main paper, comprehensive baseline comparisons for the Image Aesthetics Assessment task are presented here. Tab. 10 and Tab. 11 detail the performance of all compared methods on the TAD66K/AVA and ICAA17K/SPAQ datasets, respectively.

## F.3 Watch Time Prediction (WTP)

### F.3.1 Datasets and Experimental Setup

Three publicly available datasets and one industrial dataset are used to evaluate the proposed method. The large-scale industrial dataset (Indust. for short) is sourced from a real-world streaming short-video app with over 400 million DAUs and multi-billion impressions each day. We collect interaction logs

Table 11: The results of Image Aesthetics Assessment task on ICAA17K and SPAQ datasets.

| Method | ICAA17K | | | | SPAQ | | | |
|---|---|---|---|---|---|---|---|---|
| | MAE ↓ | XAUC ↑ | LCC ↑ | SRCC ↑ | MAE ↓ | XAUC ↑ | LCC ↑ | SRCC ↑ |
| RAPID (Lu et al., 2014) | 0.7415 | 0.6416 | 0.5164 | 0.5083 | 1.0890 | 0.6997 | 0.6565 | 0.6128 |
| AADB (Kong et al., 2016) | 0.7142 | 0.6661 | 0.5311 | 0.5195 | 1.083 | 0.7036 | 0.6646 | 0.6162 |
| PAM (Ren et al., 2017) | 0.7070 | 0.6729 | 0.5385 | 0.5247 | 1.0726 | 0.7104 | 0.6691 | 0.6222 |
| ALamp (Ma et al., 2017) | 0.6948 | 0.6847 | 0.5478 | 0.5339 | 1.0511 | 0.7250 | 0.6835 | 0.6349 |
| NIMA (Talebi & Milanfar, 2018) | 0.6957 | 0.6839 | 0.5458 | 0.5333 | 1.0756 | 0.7084 | 0.6709 | 0.6204 |
| $MP_{ada}$ (Sheng et al., 2018) | 0.6948 | 0.6848 | 0.5485 | 0.5340 | 1.0525 | 0.7240 | 0.6808 | 0.6341 |
| MLSP (Hosu et al., 2019) | 0.6814 | 0.6983 | 0.5606 | 0.5445 | 1.0428 | 0.7306 | 0.6952 | 0.6402 |
| MT-A (Fang et al., 2020) | 0.6855 | 0.6940 | 0.5558 | 0.5412 | 1.0455 | 0.7289 | 0.6862 | 0.6384 |
| BIAA (Zhu et al., 2020) | 0.6864 | 0.6932 | 0.5552 | 0.5405 | 1.0497 | 0.7259 | 0.6826 | 0.6358 |
| UIAA (Zeng et al., 2019) | 0.6889 | 0.6907 | 0.5559 | 0.5386 | 1.0469 | 0.7278 | 0.6862 | 0.6376 |
| MUSIQ (Ke et al., 2021) | 0.6740 | 0.7059 | 0.5632 | 0.5504 | 1.0427 | 0.7308 | 0.6925 | 0.6401 |
| HGCN (She et al., 2021) | 0.6813 | 0.6983 | 0.5566 | 0.5445 | 1.040 | 0.7328 | 0.6934 | 0.6417 |
| TANet (He et al., 2022) | 0.6789 | 0.7008 | 0.5599 | 0.5465 | 1.0469 | 0.7279 | 0.6844 | 0.6375 |
| MaxViT (Tu et al., 2022) | 0.6582 | 0.7227 | 0.5853 | 0.5636 | 1.042 | 0.7308 | 0.6925 | 0.6401 |
| Delegate (He et al., 2023) | 0.6345 | 0.7498 | 0.6034 | 0.5847 | 1.019 | 0.7473 | 0.7114 | 0.6545 |
| AesMamba (Gao et al., 2024) | 0.6129 | 0.7663 | 0.6137 | 0.6294 | 0.9875 | 0.7522 | 0.7261 | 0.6895 |
| **GoR with ResNet** | 0.6115 | 0.7653 | 0.6133 | 0.6305 | 0.9911 | 0.7588 | 0.7322 | 0.6886 |
| **GoR with TANet** | 0.5994 | 0.7838 | 0.6744 | 0.6675 | 0.9489 | 0.7644 | 0.7406 | 0.7189 |
| **GoR with AesMamba** | 0.5842 | 0.7913 | 0.6823 | 0.6789 | 0.8722 | 0.7648 | 0.7434 | 0.7233 |

from 2025-09-01 to 2025-09-06 and utilize the subsequent day's data for evaluation. The CIKM16[4], sourced from the CIKM16 Cup competition, is designed to predict user engagement duration in online search sessions. It contains 310,302 sessions, 122,991 items, and an average session length of 3.981. Both KuaiRand (Gao et al., 2022b) and KuaiRec (Gao et al., 2022a) are real-world datasets collected from Kuaishou app video view logs. KuaiRand comprises 26,988 users, 6,598 items, and 1,266,560 impressions, while the larger KuaiRec dataset consists of 7,176 users, 10,728 items, and 12,530,806 impressions.

Unlike traditional user behavior modeling tasks in recommendation systems, WTP does not inherently depend on history action sequences. Consequently, we employ a two-layer Multi-Layer Perceptron (MLP) as the encoder, maintaining the same configuration as the compared baseline methods.

### F.3.2 BASELINES

To demonstrate the effectiveness of our proposed approach, we comprehensively evaluate it against several state-of-the-art watch time prediction (WTP) baselines. The comparative methods are detailed as follows:

1. **VR (Value Regression):** A straightforward regression baseline that directly fits the continuous viewing duration. It is typically optimized by minimizing the Mean Squared Error (MSE) between the predicted and actual watch times.

2. **WLR** (Covington et al., 2016): This approach casts the continuous prediction task into a binary classification framework. Specifically, it assigns the observed watch time as the sample weight during the optimization of the logistic loss.

3. **D2Q** (Zhan et al., 2022): A duration-aware strategy that first partitions videos into distinct groups according to their total lengths. It then estimates a relative watch time quantile for the specific group, which is subsequently mapped back to an absolute time value to form the final prediction.

4. **CWM** (Zhao et al., 2024): Operating from a causal inference perspective, this method leverages a cost-dependent transformation to infer genuine user preferences, yielding the counterfactual watch time (CWT). The model parameters are learned by maximizing a counterfactual likelihood function constructed from observational data.

5. **TPM** (Lin et al., 2023): This technique constructs a hierarchical tree architecture to capture the dependencies among time segments at multiple resolutions. By formulating the task via ordinal regression, the expected watch time is aggregated through a weighted summation of interval lengths and their corresponding path probabilities.

---

[4]https://competitions.codalab.org/competitions/11161

6. **CREAD** (Sun et al., 2024): Building upon the principles of ordinal regression, this method introduces an adaptive mechanism to dynamically discretize continuous time into varying intervals based on error bounds. It employs sequential binary classifiers to predict the survival probability across these bins, deriving the final continuous output via expected value aggregation.

7. **SWaT** (Yang et al., 2025): Formulated upon user viewing behavioral assumptions, this statistical paradigm addresses the non-stationary nature of watch probabilities through a specialized bucketization strategy. The final duration prediction is similarly computed via probability-weighted expectations.

### F.4 LIFE TIME VALUE PREDICTION (LTV)

#### F.4.1 DATASETS AND EXPERIMENTAL SETUP

We evaluate GoR on the Criteo-SSC[5] and Kaggle[6] datasets. For both datasets, a random split of 7:1:2 is used for training, validation, and testing, respectively. Criteo-SSC is a large-scale public dataset derived from Criteo Predictive Search (CPS) logs. Each instance represents a user's click behavior, with the task being to predict conversion and associated 30-day revenue. The product price feature was excluded from the inputs. The Kaggle Dataset contains transaction records. Following (Weng et al., 2024), the task involves predicting a user's total purchase value from a specific company in the year following their initial purchase. Our experiments focus on initial purchases within 2012-03-01 and 2012-07-01, using data from the three companies with the highest transaction volume.

#### F.4.2 BASELINES

To rigorously evaluate the effectiveness of our proposed approach, we benchmark it against a variety of advanced, state-of-the-art customer lifetime value (LTV) prediction models (Drachen et al., 2018; Ma et al., 2018; Wang et al., 2019; Li et al., 2022; Liu et al., 2024; Weng et al., 2024). The specifics of these baselines are detailed as follows:

1. **Two-stage** (Drachen et al., 2018): This method splits the LTV estimation process into two distinct phases. Initially, a binary classification module determines the likelihood of user churn, which is followed by a continuous regression task to calculate the expected monetary value generated by the retained users.

2. **MTL-MSE** (Ma et al., 2018): Adopting a multi-task learning architecture, this baseline concurrently predicts the purchase conversion rate and the continuous LTV. The entire network is optimized by minimizing a standard Mean Squared Error (MSE) objective function.

3. **ZILN** (Wang et al., 2019): To address the highly skewed and long-tailed nature of LTV data, this approach models the target variable using a zero-inflated log-normal distribution. Deep neural networks are utilized to directly output the distribution parameters: the conversion probability $p$, alongside the mean $\mu$ and standard deviation $\sigma$.

4. **MDME** (Li et al., 2022): This strategy partitions the overarching LTV distribution into several distinct buckets. It first formulates a classification problem to assign instances to their respective buckets, and subsequently refines the estimation by predicting the intra-bucket residual bias, yielding a highly granular final LTV prediction.

5. **MDAN** (Liu et al., 2024): This technique categorizes users into pre-established value buckets via a multi-class classifier while concurrently learning bucket-specific embeddings through a multi-channel network. The final feature representation is constructed by aggregating these embeddings based on the classifier's probability outputs (via a weighted sum), which is then fed into the LTV regression module.

6. **OptDist** (Weng et al., 2024): Featuring a dynamic routing framework, this method adaptively assigns the most fitting sub-distribution to each user. It comprises a Distribution Learning Module (DLM) to capture diverse data patterns and a Distribution Selection Module (DSM) that dynamically routes individual customers to their optimal sub-distribution networks.

7. **HiLTV** (Xu et al., 2025): A multi-level architecture specifically designed for gaming LTV prediction. It effectively captures complex, multi-modal purchasing patterns utilizing a Zero-

---

[5]https://ailab.criteo.com/criteo-sponsored-search-conversion-log-dataset/
[6]https://www.kaggle.com/c/acquire-valued-shoppers-challenge

Inflated Mixture-of-Logistic loss, and incorporates a specialized calibration component to enhance prediction robustness for newly registered users.

For this task, we employ the same encoder architecture for GoR in Appendix F.3.1.

## F.5 ADDITIONAL RESULTS WITH MEAN AND STANDARD DEVIATION

To complement the main results, we report here the complete performance with mean and standard deviation across five runs for all datasets and tasks. Significance is assessed using paired t-tests against the strongest baseline, with improvements marked as * ($p<0.05$) and ** ($p<0.01$). These results provide a more comprehensive view of GoR's robustness and stability across domains.

Table 12: Image Aesthetics Assessment Results.

| Method | TAD66K | | | | AVA | | | |
|---|---|---|---|---|---|---|---|---|
| | MAE ↓ | XUAC ↑ | LCC ↑ | SRCC ↑ | MAE ↓ | XUAC ↑ | LCC ↑ | SRCC ↑ |
| GoR | 0.996±0.003** | 0.677±0.003** | 0.541±0.012* | 0.513±0.011** | 0.395±0.015** | 0.751±0.012* | 0.726±0.028* | 0.701±0.016** |
| Method | ICAA | | | | SPAQ | | | |
| | MAE ↓ | XUAC ↑ | LCC ↑ | SRCC ↑ | MAE ↓ | XUAC ↑ | LCC ↑ | SRCC ↑ |
| GoR | 0.5842±0.0081** | 0.7913±0.0102** | 0.6823±0.012** | 0.6789±0.0132* | 0.8722±0.0223** | 0.7648±0.0035** | 0.7434±0.0028** | 0.7233±0.0135* |

Table 13: Life Time Value Prediction Results.

| Method | Criteo-SSC | | Kaggle | |
|---|---|---|---|---|
| | MAE ↓ | SRCC ↑ | MAE ↓ | SRCC ↑ |
| GoR | 12.965±0.353** | 0.3036±0.094* | 67.075±0.112** | 0.5363±0.041** |

Table 14: Watch Time Prediction Results.

| Method | KuaiRec | | KuaiRand | | CIKM16 | |
|---|---|---|---|---|---|---|
| | MAE ↓ | XUAC ↑ | MAE ↓ | XUAC ↑ | MAE ↓ | XUAC ↑ |
| GoR | 3.194±0.031** | 0.616±0.007** | 7.032±0.132* | 0.567±0.006** | 0.808±0.019** | 0.696±0.006** |

Table 15: Facial Age Estimation Results.

| Method | UTKFace | | FG-NET | | MORPH | | CACD | |
|---|---|---|---|---|---|---|---|---|
| | MAE ↓ | CS ↑ | MAE ↓ | CS ↑ | MAE ↓ | CS ↑ | MAE ↓ | CS ↑ |
| GoR | 3.43±0.092** | 66.58±0.145** | 4.68±0.115** | 85.66±0.032** | 2.69±0.028** | 64.95±0.232** | 3.73±0.068* | 75.29±0.105** |

