# OpenReview forum: "GoR: A Unified and Extensible Generative Framework for Ordinal Regression"
_ICLR.cc/2026/Conference — ICLR 2026 Poster_

### Official Review · Reviewer_EExj · 2025-10-26

**Soundness:** 4
**Presentation:** 3
**Contribution:** 3
**Rating:** 8
**Confidence:** 5

**Summary:**

GoR reframes OR as a sequential generation task, where the model autoregressively predicts tokens representing ordinal value segments based on Coverage-Distinctiveness Index (CoDi) and Quantile-based Vocabulary. This method is novel and interesting. Although similar methods have been applied in other visual fields such as image generation and depth estimation, the contribution of applying generative models to the general ordinal regression task is commendable. Despite this paper facing a few weaknesses in qualitative analysis and missing some related works, I think the current version of the paper has reached the acceptance bar of ICLR. If the author can address my concerns, I will consider further improving my score.

**Strengths:**

1. The method is novel and interesting.

2. Experimental results show the effectiveness of the proposed method.

3. Model-agnostic framework compatible with various architectures makes the model practical and flexible.

**Weaknesses:**

1. There is a lack of analysis of the distribution of model improvements. In other words, which part of the corrections leads to the performance improvements? boundary samples, small category samples, or most samples in the whole distribution?

2. Medical disease grading is a common OR task. The medical datasets possess boundary ambiguity and long-tail problems. Testing on medical datasets can enhance the influence and persuasion of the proposed method.

3. Regarding the related work on ordinal regression, the author seems to overlook a recent development—the methods of introducing CLIP and language models, e.g, OrdinalCLIP, L2RCLIP, NumCLIP and so on.

4. Lack of experimental comparison to some popular or latest OR methods like PoE (Li et al, CVPR2021), Ord2seq (Wang et al, ICCV2023), and NumCLIP (Du et al, ECCV2024).

**Questions:**

1. This method is based on discrete modeling via vocabulary construction. What if using continuous modeling methods? Existing methods show that continuous modeling may be a better choice than discrete modeling in some fields[1].  Has the author conducted any relevant experiments?

   [1] Autoregressive Image Generation without Vector Quantization, NIPS2024.

---

> ### Author Response · Authors · 2025-11-21
>
> We sincerely appreciate your positive feedback and constructive suggestions. Below, we provide detailed responses and hope our clarifications fully address your concerns.
>
> ### **W1:**
> > There is a lack of analysis of the distribution of model improvements. In other words, which part of the corrections leads to the performance improvements? boundary samples, small category samples, or most samples in the whole distribution?
>
> That's an insightful point! We fully agree that analyzing where the model gains come from provides a much deeper understanding of GoR’s improvements. We **present analysis from both continuous-valued and discrete-valued ordinal regression tasks**.
>
> **For continuous-valued ordinal regression**, we have included a detailed interval‑based analysis exemplified by the watch‑time prediction task (Details in Sec. 3.2.2 and Appendix. F.3.3) for performance gain analysis of GoR. The resulting performance curves, along with the distribution of predicted values, are presented in Figure.5. Figure.5(a) shows that GoR outperforms the SOTA method CREAD on all watch time intervals. In addition, the predicted distribution in Figure 5(b-d) shows that GoR produces outputs that more closely match the true distribution, and it is the only method capable of predicting the value 0. This advantage comes from the flexibility of the generative formulation, which allows the model to output `<EOS>` at any step during generation. **Please refer to Sec.3.2.2 and Appendix. F.3.3 for full details**.
>
> **For discrete-valued ordinal regression**, we conduct an experiment on the UTKFace dataset for the Facial Age Estimation task. For representative age classes (e.g., 12, 51, and 83, which are randomly selected within age groups spanning young, middle-aged, and senior stages), we analyze the prediction distribution for all ground-truth samples, categorizing predictions as correct, adjacent, or other. The results reveal that while the cumulative accuracy (correct + adjacent predictions) is comparable across different methods, **our approach consistently achieves a higher proportion of correct predictions**, demonstrating GoR's enhanced discriminative power for instance ambiguity.
>
> - For age 12:
>
>     |Method|Correct|Adjacent|Other|
>     |:-:|:-:|:-:|:-:|
>     |Unimodal|53%|35%|12%|
>     |FaRL|54%|40%|6%|
>     |GoR|63%|32%|5%|
>
> - For age 51:
>
>     |Method|Correct|Adjacent|Other|
>     |:-:|:-:|:-:|:-:|
>     |Unimodal|50%|35%|15%|
>     |FaRL|52%|36%|12%|
>     |GoR|56%|34%|10%|
>
> - For age 83:
>
>     |Method|Correct|Adjacent|Other|
>     |:-:|:-:|:-:|:-:|
>     |Unimodal|36%|27%|37%|
>     |FaRL|35%|29%|36%|
>     |GoR|41%|24%|35%|
>
> ### **W2:**
> > Medical disease grading is a common OR task. The medical datasets possess boundary ambiguity and long-tail problems. Testing on medical datasets can enhance the influence and persuasion of the proposed method.
>
> Thanks for the suggestion! Medical disease/physiological grading is an important application area of ordinal regression, such as Bone Age Assessment, Diabetic Retinopathy Grading, and Glaucoma Grading. Following your suggestion, we add an additional experiment in a medical scenario.
> Specifically, **we follow the experimental setup of SupReMix [1] and evaluate GoR on the RSNA Bone Age Dataset**. This task aims to predict bone age from X-ray images (0–228 months), representing a typical ordinal regression problem with pronounced long-tail distribution and ambiguous boundaries between adjacent classes. The results show that GoR significantly outperforms SupReMix and other SOTA methods in terms of MAE metric, demonstrating its effectiveness for medical ordinal prediction. We plan to extend GoR to more medical datasets in future work.
>
> | Method   | MAE |
> |----------|---------|
> | SIMCLR   | 8.28    |
> | SupCon   | 6.79    |
> | AdaCon   | 6.04    |
> | RnC      | 5.30    |
> | SupReMix | 4.08 |
> | **GoR** | **3.88** |
>
> **References:**
>
> [1] SupReMix: Supervised Contrastive Learning for Medical Imaging Regression with Mixup, Arxiv'25
>
> ### **W3:**
> > Regarding the related work on ordinal regression, the author seems to overlook a recent development—the methods of introducing CLIP and language models, e.g, OrdinalCLIP, L2RCLIP, NumCLIP and so on.
>
> Thanks for pointing this out! Recent works that incorporate CLIP and language models into ordinal regression (e.g., OrdinalCLIP, L2RCLIP, NumCLIP) indeed represent a rapidly growing and promising research direction. These methods typically align image features with textual ordinal descriptions to enhance the semantic understanding and generalization of ordinal relationships. **We have added a discussion and citations of these methods in the updated Related Work section (highlighted in blue)**.

---

> ### Author Response · Authors · 2025-11-21
>
> ### **W4:**
> > Lack of experimental comparison to some popular or latest OR methods like PoE (Li et al, CVPR2021), Ord2seq (Wang et al, ICCV2023), and NumCLIP (Du et al, ECCV2024).
>
> Thanks for your helpful comment. For PoE and Ord2Seq, we have already included the comparisons under the HIC task on Appendix. F.6 of the main paper, and the methodological differences between GoR and Ord2Seq are discussed in detail in the Related Work section (**we have highlighted in line 860**).
> In the revision, we additionally include the PoE results on Facial Age Estimation and Image Aesthetics Assessment (see Tables 3, 8, and 9, where the new results are highlighted). All evaluations follow the protocol in Appendix F.1.3.
>
> As for NumCLIP and other CLIP-based approaches, their reliance on visual inputs makes them unsuitable for many non-visual OR scenarios (e.g., watch-time prediction, LTV estimation, bidding strategy optimization, or ordinal time-series prediction). In contrast, GoR targets a modality-agnostic, task-general ordinal regression framework. Moreover, they inevitably rely on strong vision–language pretraining priors, which leads to **a fundamentally different problem setting from GoR that is trained without using external pretraining data**. Therefore, **a direct comparison would not be entirely fair**. Instead, we provide a systematic discussion of these methods in the Related Work section rather than treating them as primary baselines (updates highlighted in blue).
>
> ### **Q1:**
> > This method is based on discrete modeling via vocabulary construction. What if using continuous modeling methods? Existing methods show that continuous modeling may be a better choice than discrete modeling in some fields[1]. Has the author conducted any relevant experiments?
>
> Insightful point! **We fully agree that continuous generative modeling is a promising direction for OR**.
>
> Discrete modeling of GoR via vocabulary construction and continuous generative modeling (e.g., diffusion, MAR you cited) represent two distinct generative paradigms.
> Our motivation for proposing GoR stems from the limitations of the widely adopted rank-based methods under the Continuous Space Discretization paradigm, which suffer from bin-wise prediction independence and rigid interval boundaries (lines 50–74). GoR addresses these issues by modeling sequential ordinal dependencies via dynamic `〈EOS〉`-terminated token generation, offering adaptive resolution and interpretable step-wise refinement.
>
> Based on our investigation, we have not found any existing work that applies DDPM or Normalizing Flows to continuous ordinal regression. The only related generative work is [1]. However, it is essentially a rank-based ordinal classification method rather than continuous generative modeling. Specifically, it uses a diffusion architecture to model binary (0/1) probabilities under a fixed interval partition, similar to CREAD discussed in line 1236. Thus, it is essentially a discrete modeling method rather than true continuous modeling.
>
> **Regarding continuous generative modeling for OR, in fact, we have already conducted preliminary experiments to explore this possibility**.
> Specifically, we model ordinal regression as a continuous generative problem using a simple VAE architecture, where the objective is to reconstruct the continuous ordinal target on the watch-time prediction task (named VAE-demo).  Interestingly, we observed a noticeable improvement in MAE compared to existing SOTAs such as CREAD.
>
> | Method | MAE | XAUC |
> |:--:|:--:|:--:|
> | TPM | 3.456 | 0.571  |
> | SWaT | 3.438 | 0.585  |
> | CREAD | 3.307 | 0.594  |
> | VAE-demo| 3.246 | 0.606 |
> | GoR | 3.194 | 0.616 |
>
> We conjecture that continuous generative models, by explicitly modeling the data distribution, may implicitly encode ordinal relations in the latent space. While this is a different generative modeling path from GoR, we view continuous generative modeling for OR as a valuable direction for further investigation, and we are actively exploring it. We would be very happy to engage in further discussion if you have thoughts or suggestions!
>
> **References:**
>
> [1] Parameterized Diffusion Optimization enabled Autoregressive Ordinal Regression for Diabetic Retinopathy Grading, MICCAI 2025.
>
> If you have any further questions or would like to discuss any aspect in more detail, please feel free to contact us at any time. We would be more than happy to provide additional clarification or engage in further discussion.

---

### Official Review · Reviewer_yYd1 · 2025-10-29

**Soundness:** 2
**Presentation:** 4
**Contribution:** 3
**Rating:** 8
**Confidence:** 4

**Summary:**

This paper introduces GoR (Generative Ordinal Regression), a novel generative framework that formulates ordinal regression as an autoregressive token generation task, terminating with a dynamic <EOS> token. Instead of relying on traditional continuous space discretization or rank-based classification approaches, GoR represents the target ordinal value as a sequence of additive segments drawn from a learned vocabulary. To support this framework, the authors derive a bias-variance decomposed MSE error bound and propose Coverage–Distinctiveness index for principled vocabulary construction, addressing the trade-off between quantization bias and statistical variance. The framework is model-agnostic, supporting a wide range of encoders and decoders, and is extensible to standard generative learning techniques. Extensive experiments show that GoR achieves state-of-the-art performance consistently.

**Strengths:**

1. The proposal to model ordinal regression as sequential generative modeling is a novel paradigm shift.
2. The bias-variance decomposition provides theoretical grounding for vocabulary design.
3. Good potential for future works as it is compatible with standard training techniques used in generative modeling
4. Achieves consistent gains on the benchmarks.

**Weaknesses:**

1. Decoding time grows linearly with the number of tokens, which itself varies with the resolution and magnitude of the target ordinal value. The author should consider to include efficiency evaluation.
2. No comparisons with other generative-based models like DDPM or Normalizing Flows in continuous ordinal prediction.
3. Longer sequences amplify token-level noise due to accumulating prediction errors across steps. Prediction accuracy may degrade on long sequences or in fine-resolution tasks, especially under greedy decoding. The authors should consider to analyse the trade-off of the model.

**Questions:**

See Weaknesses Above.

---

> ### Author Response · Authors · 2025-11-21
>
> We are grateful for your positive feedback and the thoughtful evaluation of our work! Below, we offer detailed responses and hope our clarifications further address your concerns.
>
> ### **W1/Q1:**
> > Decoding time grows linearly with the number of tokens, which itself varies with the resolution and magnitude of the target ordinal value.
>
> Thanks for your insightful comment! This concern is effectively addressed under our systematic vocabulary construction mechanism and label decomposition strategy.
>
> Specifically, the initial vocabulary is adaptively constructed based on the true label distribution of each dataset: we employ an iterative procedure using a fixed percentile $q$ to select tokens of appropriate magnitudes for different value ranges, ensuring that each label value is matched with a token of a compatible scale and preventing unnecessarily long sequences caused by decomposing large values into many small ones (details in Alg. 2 of the main paper). Subsequently, the CoDi-based pruning further balances token usage while preserving representational fidelity, resulting in a more compact and evenly utilized vocabulary.
> **Under this mechanism, even when label ranges span several orders of magnitude, large values are still decomposed through appropriately scaled tokens, so the sequence length does not grow proportionally with label magnitude.**
>
> For example, in the watch-time prediction task, the vocabulary is
>
> ```[634632, 153653, 90157, 61751, 47414, 38653, 32973, 28976, 25994, 23736, 21926, 20449, 19183, 18059, 17067, 16179, 15352, 14599, 13910, 13273, 12689, 12143, 11641, 11171, 10714, 10267, 9828, 9387, 8943, 8491, 8040, 7582, 7127, 6670, 6202, 5712, 5192, 4629, 4020, 3396, 2779, 2204, 1477, 1260, 897, 647, 541, 478, 438, 410, 381, 352, 324, 295, 266, 237, 209, 181, 153, 124, 96, 70, 49, 32, 26, 23, 20, 17, 15, 13, 11, 9, 7, 5, 3, 1] (unit: ms)```
>
> For two labels 250,000 and 5,000—although the former is fifty times larger—their decomposed sequences differ only slightly in length:
>
> + 250,000 → [153653, 61751, 28976, 5192, 410, 15, 3]  → length 7
> + 5,000 → [4629, 352, 17, 1, 1]  → length 5
>
> This result demonstrates that GoR **maintains controlled sequence lengths within a dataset, ensuring that decoding complexity does not deteriorate as target values grow**.
>
> > The author should consider to include efficiency evaluation.
>
> Following your suggestion, we provide an efficiency evaluation on the watch-time prediction task using the KuaiRec dataset. We compare GoR with two representative ordinal regression baselines:
>
> (1) CREAD: A classical rank-based method that discretizes the label range into a sequence of bins and models a binary classifier for each bin to predict whether the watch time exceeds the corresponding threshold.
>
> (2) TPM: A tree-structured approach that performs hierarchical ordinal decisions. It recursively determines which sub-interval the label falls into until reaching a leaf node.
>
> Below we report both performance and inference time on the full test set of **4,676,570 samples**:
>
> | Method | MAE | XAUC | Inference Time |
> |:--:|:--:|:--:|:--:|
> | TPM | 3.456 | 0.571 | 4 min 15 s |
> | CREAD | 3.307 | 0.594 | 2 min 16s |
> | GoR | 3.194 | 0.616 | 5 min 03 s |
>
> On the full test set of 4.67M samples, GoR is only slightly slower than CREAD (by less than 3 minutes) and TPM (by less than 1 minute), while outperforming all baselines by a clear margin. **Furthermore, GoR has already been successfully deployed in industrial-scale recommendation systems with stringent real-time requirements**, and achieves a +0.112% increase in app usage time (p = 0.01) in A/B tests on a real-world platform with over 400 million DAUs (Appendix F.3.4). This real-world deployment demonstrates that GoR meets production-level latency constraints and offers a practically acceptable latency–performance trade-off.

---

> ### Author Response · Authors · 2025-11-21
>
> ### **W2/Q2:**
> > No comparisons with other generative-based models like DDPM or Normalizing Flows in continuous ordinal prediction.
>
> Thank you for your valuable suggestion. Based on our investigation, **we have not found any existing work that applies DDPM or Normalizing Flows to continuous ordinal regression**, and therefore no direct comparison can be provided.
> The only related generative work is [1]. However, it is essentially a rank-based ordinal classification method rather than continuous generative modeling. Specifically, it uses a diffusion architecture to model binary (0/1) probabilities under a fixed interval partition, similar to CREAD discussed in line 1236. Thus, it is essentially a discrete modeling method rather than true continuous modeling.
>
> In contrast to diffusion-based continuous modeling, GoR is a discrete generative paradigm analogous to language models. It explicitly models ordinal dependencies through a sequence of tokens and adaptively terminates generation via the `<EOS>` token, directly alleviating the interval-independence and rigid-boundary issues that dominate existing OR methods.
>
> We agree that exploring continuous generative models (e.g., diffusion or flows) for continuous ordinal regression is a promising future direction. However, bridging the gap between such continuous modeling frameworks and the specific structural requirements of ordinal regression will require additional theoretical and architectural developments. We therefore leave this avenue for future work.
>
> [1] Parameterized Diffusion Optimization enabled Autoregressive Ordinal Regression for Diabetic Retinopathy Grading, MICCAI 2025.
>
> ### **W3/Q3:**
> > Longer sequences amplify token-level noise due to accumulating prediction errors across steps. Prediction accuracy may degrade on long sequences or in fine-resolution tasks, especially under greedy decoding. The authors should consider to analyse the trade-off of the model.
>
> That's a good question! Autoregressive models are indeed susceptible to cumulative errors, particularly when long decoding paths amplify token-level noise. While GoR is also a generative autoregressive framework, this issue is effectively mitigated as follows:
>
> 1. **Reasonable sequence length**
>
>     As described in Q1, GoR’s vocabulary construction and label decomposition strategy ensure that each label value is matched with a token of comparable magnitude. As a result, even extremely large targets are decomposed into compact sequences, whose lengths remain stable across samples. For example, in the watch-time prediction task,
>     labels 250,000 and 5,000 differ by 50× in scale but yield sequences only two steps apart (details in Q1), demonstrating that **sequence length does not scale with value magnitude**. This inherently shortens the error accumulation path.
>
> 2. **Effective optimization**
>
>     As discussed in Sec. 3.2.4, **one major advantage of GoR is its compatibility with established optimization techniques for autoregressive language models without requiring additional parameters**, including Curriculum Learning, N-gram Prediction, DPO-based Reinforcement Learning and so on. These strategies effectively reduce exposure bias and stabilize sequential prediction, as verified in our experiments in Sec. 3.2.4.
>
> 3.  **Trade-off analysis through decoding-length ablation**
>
>     We further analyze the trade-off on the watch-time prediction task by varying the maximum decoding length while keeping the vocabulary fixed (Details refer to the answer of Q2 for reviewer n5TF). The results show:
>     + overly small decoding length severely degrades performance
>     + performance improves with longer allowed sequences
>     + it stabilizes near the maximum length observed during training
>     This confirms that **these strategies enable GoR to better approximate ordinal values progressively and accurately through its sequential generation process**.
>
>         |Max Decoding Length | MAE    | XAUC   |
>         |:--:|:--:|:--:|
>         | 1 | 8.6126 | 0.5335 |
>         | 2 | 6.1023 | 0.5681 |
>         | 4 | 4.3125 | 0.5804 |
>         | 8 | 3.3122 | 0.5953 |
>         | 16 | 3.2162 | 0.6111 |
>         | Training Max Length = 23 | 3.1942 | 0.6164 |
>         | 30 | 3.2058 | 0.6136 |
>
> Therefore, GoR effectively mitigates the impact of error accumulation and exposure bias. Moreover, the moderate sequential generation process is a key contributor to its strong predictive performance.
>
> If you have any further questions or would like to discuss any aspect in more detail, please feel free to contact us at any time. We would be more than happy to provide additional clarification or engage in further discussion.

---

### Official Review · Reviewer_BLVB · 2025-10-31

**Soundness:** 3
**Presentation:** 3
**Contribution:** 3
**Rating:** 6
**Confidence:** 2

**Summary:**

This paper introduces a novel generative framework termed GoR for ordinal regression that reformulates scalar ordinal prediction as autoregressive token generation. By explicitly modeling ordinal dependencies and dynamically controlling prediction granularity via a learnable ⟨EOS⟩ token, the method overcomes the rigidity and boundary ambiguity in conventional discretization- and ranking-based approaches. The proposed CoDi-guided vocabulary construction is theoretically grounded through a bias–variance decomposed MSE bound, ensuring both representational flexibility and cross-domain adaptability. Extensive experiments across diverse tasks demonstrate consistent and substantial improvements over strong baselines, highlighting GoR as a promising and unified paradigm for future research in ordinal modeling.

**Strengths:**

-The sequential generation with a dynamic ⟨EOS⟩ explicitly models ordinal dependencies and enables coarse-to-fine refinement, which can offer strong interpretability and flexibility.

-The paper delivers rigorous analysis of limitations in rank-based methods and a principled MSE error bound, which provides solid justification for the proposed approach.

-Extensive experiments across diverse domains show clear and consistent improvements over strong baselines, thereby demonstrating strong generalizability.

**Weaknesses:**

-It remains unclear whether the improvements can be mainly attributed to the proposed paradigm or simply from the stronger autoregressive decoder; further controlled ablations are needed.

-The effectiveness of the proposed method depends on empirical tuning, which may limit robustness across tasks.

-The sequential generation introduces longer prediction paths and potentially higher latency, which poses concerns for efficiency and industrial deployment.

-Autoregressive decoding may suffer from compounding errors. The paper does not sufficiently address exposure bias, nor analyze the trade-off between robustness gains from beam search and increased inference cost.

-The sequence length varies with label range and distribution, potentially affecting stability and efficiency. More evidence is needed to verify consistent performance across tasks with diverse label scales.

**Questions:**

-It remains unclear whether the improvements can be mainly attributed to the proposed paradigm or simply from the stronger autoregressive decoder; further controlled ablations are needed.

-The effectiveness of the proposed method depends on empirical tuning, which may limit robustness across tasks.

-The sequential generation introduces longer prediction paths and potentially higher latency, which poses concerns for efficiency and industrial deployment.

-Autoregressive decoding may suffer from compounding errors. The paper does not sufficiently address exposure bias, nor analyze the trade-off between robustness gains from beam search and increased inference cost.

-The sequence length varies with label range and distribution, potentially affecting stability and efficiency. More evidence is needed to verify consistent performance across tasks with diverse label scales.

---

> ### Author Response · Authors · 2025-11-21
>
> We are grateful for your positive feedback and the thoughtful evaluation of our work! Below, we offer detailed responses and hope our clarifications further address your concerns.
>
> ### **Q1/W1:**
>
> > It remains unclear whether the improvements can be mainly attributed to the proposed paradigm or simply from the stronger autoregressive decoder; further controlled ablations are needed.
>
> Thanks for your thoughtful comment. To clarify the source of GoR’s performance improvements, we conducted three ablation experiments:
>
> 1. Ablation across different autoregressive decoders:
>
>     As shown in Figure. 4 of the main paper, we evaluated RNN, GRU, LSTM, and Transformer decoders. The results show that **even the simplest RNN-based GoR significantly outperforms existing SOTA methods**, demonstrating that the gains are not dependent on decoder sophistication and validating the model-agnostic nature of GoR (details in line 407).
>
> 2. Parameter-controlled comparison:
>
>     We standardized the parameter count of baseline SOTA models of the watch time prediction task to match GoR. **Under equal parameter sizes, GoR consistently surpasses SOTA methods across all metrics**, indicating that the performance gains arise from the proposed paradigm rather than scale. This experiment has been added to the revision in Appendix F.3.4, with the updates highlighted in blue.
>
>     | Method | Parameters | MAE | XAUC |
>     |:--:|:--:|:--:|:--:|
>     | VR | 0.86M      | 7.634 | 0.534  |
>     | TPM | 0.86M      | 3.456 | 0.571  |
>     | CREAD | 0.86M      | 3.307 | 0.594  |
>     |SWaT | 0.88M      | 3.438 | 0.585  |
>     | VR-large    | 4.34M      | 7.556 | 0.545  |
>     | TPM-large   | 4.34M      | 3.432 | 0.577  |
>     | CREAD-large | 4.34M      | 3.293 | 0.599  |
>     |SWaT-large | 4.35M      | 3.406 | 0.591  |
>     | **GoR (ours)** | **4.18M** | **3.194** | **0.616** |
>
> 3. Ablation on the decoding length:
>
>     Following reviewer n5TF’s suggestion, we fixed the vocabulary and varied only the maximum decoding length on the watch-time prediction task. When the maximum length is severely restricted (e.g., 1), performance drops significantly. As the decoding length limit increases, performance rises progressively and then stabilizes. This pattern confirms that GoR benefits from modeling ordinal dependencies via conditional generation and progressively refining the prediction from coarse to fine (Details refer to the answer of Q2 for reviewer n5TF).
>
>     |Max Decoding Length | MAE    | XAUC   |
>     |:--:|:--:|:--:|
>     | 1 | 8.6126 | 0.5335 |
>     | 2 | 6.1023 | 0.5681 |
>     | 4 | 4.3125 | 0.5804 |
>     | 8 | 3.3122 | 0.5953 |
>     | 16 | 3.2162 | 0.6111 |
>     | Training Max Length = 23 | 3.1942 | 0.6164 |
>     | 30 | 3.2058 | 0.6136 |
>
> Together, these results show that: (1) the gains are independent of the specific decoder architecture, (2) they do not arise from increased parameter count, and (3) **the generative sequential modeling itself contributes substantially to performance improvements**. Therefore, the improvements of GoR are primarily driven by the proposed generative ordinal modeling paradigm rather than by a stronger autoregressive decoder.
>
> ### **Q2/W2:**
>
> >  The effectiveness of the proposed method depends on empirical tuning, which may limit robustness across tasks.
>
> Thank you for raising this insightful question. We respectfully clarify that GoR is designed to minimize dependence on empirical tuning, achieving strong robustness across tasks through adaptive mechanisms rather than manual adjustment. **This robustness primarily comes from our systematic vocabulary construction and label decomposition strategy**, which allows the model to automatically adapt to the label scales and distribution characteristics of each domain/dataset and construct an appropriate vocabulary, thereby greatly reducing the need for empirical tuning.
>
> Specifically, the initial vocabulary is adaptively constructed based on the true label distribution of each dataset: we employ an iterative procedure using a fixed percentile $q$ to select tokens of appropriate magnitudes for different value ranges, ensuring that each label value is matched with a token of a compatible scale and preventing unnecessarily long sequences (details see Alg.2 of main paper). Subsequently, the CoDi-based pruning further balances the token usage while preserving representational fidelity, reducing the model’s sensitivity to task-specific hyperparameter tuning.
>
> Empirically, we observe that GoR’s key hyperparameters remain highly stable across tasks. For example, the optimal value of $\beta$ is consistently around 0.7. Importantly, **even without any task-specific tuning, GoR achieves state-of-the-art performance on 17 benchmarks spanning 6 domains**, illustrating the robustness of the framework and the reliability of its default configuration. Additional tuning may bring marginal improvements, but it is not required for GoR to achieve its strong performance.

---

> ### Author Response · Authors · 2025-11-21
>
> ### **Q3/W3:**
> > The sequential generation introduces longer prediction paths and potentially higher latency, which poses concerns for efficiency and industrial deployment.
>
> Thanks for your valuable comment. Since autoregressive models generate predictions step by step, GoR naturally introduces an efficiency–performance trade-off, which we have discussed in Appendix. G of the main paper.
>
> Crucially, as discussed in Appendix F.3.4, **GoR has already been deployed in industrial-scale recommendation systems with stringent real-time constraints**, achieving a +0.112% lift in app usage time (p = 0.01) in A/B tests on a commercial platform with over 400M DAUs. This real-world deployment demonstrates that GoR **meets production-level latency constraints and offers a practically acceptable latency–performance trade-off**.
>
> Moreover, our systematic **vocabulary construction and label decomposition strategy ensures that sequence length does not grow proportionally with label magnitude**, even when labels span several orders of magnitude (Details refer to the answer Q5 of Reviewer BLVB). This mechanism effectively prevents unnecessary growth in decoding steps and avoids excessive latency for large values.
>
> Similar to the early challenges faced by initial Transformer models with non-linear time complexities, we would like to point out that GoR achieves SOTA results across 17 diverse OR datasets spanning six domains, which represents a foundational step toward a more powerful generative paradigm. Furthermore, we believe that, as a novel and powerful baseline for the generative paradigm, GoR will catalyze the development of even more efficient methods, such as non-autoregressive generative modeling, which we leave open for the future.
>
> ### **Q4/W4:**
>
> > Autoregressive decoding may suffer from compounding errors. The paper does not sufficiently address exposure bias.
>
> Thanks for your thoughtful comment. Autoregressive decoding inherently suffers from compounding errors and exposure bias, which are widely recognized issues in language modeling of the NLP field. As a general generative sequence modeling paradigm, GoR also encounters similar challenges.
>
> However, as discussed in Sec. 3.2.4, **one major advantage of GoR is its ability to seamlessly incorporate existing optimization strategies for generative language models**. We have already included several relevant techniques in our experiments, including Curriculum Learning, N-gram Prediction, and DPO-based Reinforcement Learning. The ablation results in Table.5 of the main paper show these strategies **effectively mitigate error accumulation and exposure bias and are achieved without introducing additional model parameters**.
>
> > analyze the trade-off between robustness gains from beam search and increased inference cost
>
> Regarding beam search, GoR is fully compatible with this inference strategy, which can play two roles:
>
> (1) It can directly serve as an inference method to improve predictive performance, but can introduce noticeable inference latency. The table below shows the performance–efficiency trade-off of beam search, reporting the total inference time on the **4,676,570** test samples of the KuaiRec dataset and the corresponding per-second processing speed.
>
> | Beam Size | MAE | XAUC | Inference Time (Total) | Speed (items/s) |
> |:--:|:--:|:--:|:--:|:--:|
> | 1 (Greedy) | 3.194 | 0.616 | 303s | 15434 |
> | 2         | 3.190 | 0.618 | 520s |  8,993 |
> | 4         | 3.186 | 0.622 | 880s |  5,314 |
> | 8         | 3.184| 0.624 | 1,650s |  2,834 |
> | 16        | 3.180 | 0.625 | 3,200s |  1,461 |
>
> (2) It can provide positive and negative candidate samples for DPO post-training (results shown in the last row of Table.5 in the main paper).
>
> However, as stated in line 485 of the main paper, GoR with the efficient configuration in row (e) of Table 5 already surpasses existing SOTA methods by a large margin across tasks. **To avoid the considerable computational and time cost of RL-based post-training and beam-search inference, we do not include these components in the reported results**.

---

> ### Author Response · Authors · 2025-11-21
>
> ### **Q5/W5:**
>
> > The sequence length varies with label range and distribution, potentially affecting stability and efficiency.
>
> Thanks for your valuable comment. This concern is effectively addressed under our systematic vocabulary construction mechanism and label decomposition strategy.
>
> Specifically, the initial vocabulary is adaptively constructed based on the true label distribution of each dataset: we employ an iterative procedure using a fixed percentile $q$ to select tokens of appropriate magnitudes for different value ranges, ensuring that each label value is matched with a token of a compatible scale and preventing unnecessarily long sequences caused by decomposing large values into many small ones (details in Alg. 2 of the main paper). Subsequently, the CoDi-based pruning further balances token usage while preserving representational fidelity, resulting in a more compact and evenly utilized vocabulary.
> **Under this mechanism, even when label ranges span several orders of magnitude, large values are still decomposed through appropriately scaled tokens, so the sequence length does not grow proportionally with label magnitude.**
>
> For example, in the watch-time prediction task, the vocabulary is
>
> ```[634632, 153653, 90157, 61751, 47414, 38653, 32973, 28976, 25994, 23736, 21926, 20449, 19183, 18059, 17067, 16179, 15352, 14599, 13910, 13273, 12689, 12143, 11641, 11171, 10714, 10267, 9828, 9387, 8943, 8491, 8040, 7582, 7127, 6670, 6202, 5712, 5192, 4629, 4020, 3396, 2779, 2204, 1477, 1260, 897, 647, 541, 478, 438, 410, 381, 352, 324, 295, 266, 237, 209, 181, 153, 124, 96, 70, 49, 32, 26, 23, 20, 17, 15, 13, 11, 9, 7, 5, 3, 1] (unit: ms)```
>
> For two labels 250,000 and 5,000—although the former is fifty times larger—their decomposed sequences differ only slightly in length:
> + 250,000 → [153653, 61751, 28976, 5192, 410, 15, 3]  → length 7
> + 5,000 → [4629, 352, 17, 1, 1]  → length 5
>
> This result demonstrates that GoR **maintains controlled sequence lengths within a dataset, ensuring that decoding complexity does not deteriorate as target values grow**.
>
> > More evidence is needed to verify consistent performance across tasks with diverse label scales.
>
> We would like to point out that the vocabulary in GoR is not constructed once and then reused across all tasks. Instead, **it is adaptively built for each dataset according to its label range, scale and distribution**. In other words, datasets from different domains obtain their own specific vocabularies and sequence representations, all following the unified principles of GoR’s systematic vocabulary construction and label decomposition strategy.
>
> Under this mechanism:
> + **Within a dataset**, extremely large labels can always be decomposed with suitably scaled tokens, keeping sequence-length variation controlled even when label ranges are wide.
> + **Across datasets**, the adaptive vocabulary naturally adjusts to different label ranges and scales.
>
> Concretely, GoR maintains stable performance across tasks with widely varying label scales, including:
> + small-range, near-continuous labels (Image Aesthetic Assessment)
> + small-range, discrete labels (Historical Image Dating)
> + large-range, discrete integer labels (Facial Age Estimation)
> + large-range, continuous and long-tailed labels (Lifetime Value, Watch Time Prediction, Real-Time Bidding Bid Shading)
>
> GoR **achieves SOTA performance across all these diverse label distributions and scales**, demonstrating its stability, efficiency, and cross-domain robustness.
>
> If you have any further questions or would like to discuss any aspect in more detail, please feel free to contact us at any time. We would be more than happy to provide additional clarification or engage in further discussion.

---

### Official Review · Reviewer_n5TF · 2025-11-01

**Soundness:** 2
**Presentation:** 3
**Contribution:** 2
**Rating:** 4
**Confidence:** 4

**Summary:**

This paper proposes a new method for ordinal regression that treats it as a sequence generation task, predicting segments step by step until an END token. It avoids issues in traditional methods like boundary ambiguity and rigid binning, and introduces a way to balance bias and variance using a new metric called CoDI. It works well across many datasets and is easy to plug into existing generative models.

**Strengths:**

- This paper is well-written and easy to follow.
- The experiments show consistent improvements over SOTA across 17 datasets in 6 domains

**Weaknesses:**

- Is it possible that vocabulary pruning may lead to loss of information, especially affecting minority classes in imbalanced datasets?
- Does GoR support control over output sequence length, or does it only rely on detecting <EOS>? If no, how to control the maximum length of the output sequence, and will it take a longer time for inference? If yes, what would be the performance when limiting the maximum length of the output sequence to a certain value, e.g., 1? What are the impacts on inference time and performance if the sequence length is limited? More evaluations should be included to demonstrate.
- How was the threshold $\epsilon$ and percentage $\beta$ determined and are there any effects on the final results?

**Questions:**

See Weaknesses.

---

> ### Author Response · Authors · 2025-11-21
>
> Thanks for your valuable comments. We provide detailed responses below and hope that our clarifications would address your concerns.
>
> ### **W1/Q1:**
>
> > Is it possible that vocabulary pruning may lead to loss of information?
>
> CoDi-based pruning does not cause information loss, but instead performs a beneficial reduction of redundancy that improves generalization.
> This is because CoDi prunes tokens **not solely based on frequency**, but through a principled trade-off between:
>
> + Coverage: Quantifies the usage frequency of a token
>
> + Distinctiveness: Measures the irreplaceability of a token in representing certain values
>
> Tokens associated with minority or long-tail regions may have low coverage, but if they exhibit high distinctiveness, meaning no other tokens can effectively replace them, they may not be pruned. This Coverage-Distinctiveness trade-off is supported by both theoretical and empirical evidence:
>
> + Theorem 1 in the main paper shows that overly large vocabularies can increase variance and CoDi pruning within an appropriate range can improve the bias–variance trade-off while preserving ordinal information.
>
> + Performance improvement: Figure. 6(b) of the main paper demonstrates that moderate vocabulary pruning based on CoDi **leads to noticeable performance gains**.
>
> Therefore, CoDi selectively prunes redundant and replaceable tokens, minimizing the risk of removing meaningful information and providing a beneficial reduction of redundancy that improves overall performance.
>
> > especially affecting minority classes in imbalanced datasets?
>
> In our design, vocabulary pruning does not harm minority classes. Instead, it reduces token-level imbalance. This comes from two complementary mechanisms:
>
> 1. **Initial vocabulary construction already alleviates imbalance.**
>
>     As described in Appendix E, the initial vocabulary is constructed using an iterative percentile-based strategy. At each step, we select a token corresponding to the q-th percentile of the label distribution, subtract it to form residuals, and repeat until residuals are negligible. This design **promotes a balanced token usage from the outset and lays a solid foundation for subsequent pruning**.
>
> 2. **CoDi pruning further balances token usage.**
>
>     As explained above, CoDi removes only redundant tokens by jointly considering Coverage and Distinctiveness to further balance token usage. On the imbalanced long-tailed KuaiRec dataset, Figure. 6(a) of the main paper shows that pruning smooths the token frequency distribution and alleviates long-tail effects.
>
> As a summary, both the initial vocabulary construction and the subsequent CoDi pruning **promote balanced token usage**, alleviating token-level imbalance and preventing dominance by head tokens. The **pruning acts as a beneficial reduction of redundancy** that improves generalization, ultimately enhancing performance without harming minority-class predictions.

---

> ### Author Response · Authors · 2025-11-21
>
> ### **W2/Q2:**
> > Does GoR support control over output sequence length, or does it only rely on detecting `<EOS>`?
>
> Similar to large language models in NLP, GoR's sequence generation relies on the `<EOS>` token to determine termination while **also allowing the inference length to be constrained by setting a maximum step limit**. The maximum step limit is set to the maximum sequence length observed in the training data to avoid abnormally long sequences and control inference cost.
>
> > If yes, what would be the performance when limiting maximum length of output sequence to a certain value, e.g., 1?
>
> Following your advice, we conduct an ablation study on the watch-time prediction task, varying only the maximum decoding length during inference with the vocabulary fixed. The results (table below) show that when the length is constrained to an extremely small value (e.g., 1), the performance drops significantly. As the limit increases, **the performance steadily improves and stabilizes when approaching the maximum sequence length seen during training**.
>
> This trend indicates that GoR explicitly models sequential ordinal dependencies through conditional generation and progressively refines predictions from coarse to fine, thereby validating the motivation behind our generative paradigm.
>
> |Max Decoding Length|MAE|XAUC|Inferenece Time (Total)|Speed (items/s)|
> |:-:|:-:|:-:|:-:|:-:|
> |1|8.6126|0.5335|147s|31,813|
> |2|6.1023|0.5681|154s|30,367|
> |4|4.3125|0.5804|168s|27,837|
> |8|3.3122|0.5953|194s|24,106|
> |16|3.2162|0.6111|255s|18,339|
> |Training Max Length=23|3.1942|0.6164|303s|15,434|
> |30|3.2058|0.6136|350s|13,362|
>
> > What are the impacts on inference time and performance if the sequence length is limited?
>
> The table reports both the total inference time of **4,676,570** test samples and the per-second processing speed on the KuaiRec dataset.
> As shown, **smaller decoding limits shorten inference time but severely degrade performance, whereas larger limits incur slightly higher latency yet enable GoR to realize its full sequential-refinement capability and reach better performance.**
> The trend is not strictly linear because the encoder cost is fixed across all settings, and many samples finish decoding within only a few steps, so increasing the maximum length does not necessarily increase their actual decoding time.
>
> Similar to other autoregressive models that generate predictions step by step, limiting GoR's decoding length naturally introduces an efficiency–performance trade-off, which we have discussed in the Appendix. G of the main paper.
> Crucially, our systematic vocabulary construction and label decomposition strategy ensure that sequence length does not grow proportionally with label magnitude, even when labels span several orders of magnitude (Details refer to answer Q5 of Reviewer BLVB). This design prevents excessive latency for large targets, and setting the maximum decoding length to maximum sequence length observed during training provides a practical and balanced efficiency–performance trade-off.
>
> Moreover, as discussed in Appendix F.3.4, GoR has already been deployed in industrial-scale recommendation systems with stringent real-time constraints, achieving a +0.112% lift in app usage time (p = 0.01) in A/B tests on a commercial platform with over 400M DAUs. This real-world deployment demonstrates that GoR meets production-level latency constraints and offers a practically acceptable latency–performance trade-off.
> ### **W3/Q3:**
> > How was threshold $\epsilon$ and $\beta$ percentage determined?
>
> Both parameters are selected through ablation studies on final performance, and the analysis is provided below.,
> > Are there any effects on the final results?
>
> As described in line 234, the threshold $\epsilon$ controls the reconstruction precision of the Ordinal Target Sequencing process, ensuring that the generated sequential label accurately reconstructs or approximates the ground truth value. To balance reconstruction fidelity and computational feasibility, $\epsilon$ is typically set to a small value. We evaluate several settings (0.01, 0.005, 0.001), and the results **show negligible differences** (see the table below). We therefore adopt 0.005 as the default configuration (line 1117).
>
> |$\epsilon$|MAE|XAUC|
> |:--:|:--:|:--:|
> |0.01|3.1953|0.6138|
> |0.005|3.1942|0.6164|
> |0.001|3.1942|0.6163|
>
> The percentage $\beta$ denotes the retained vocabulary ratio, which controls the CoDi pruning strength. A detailed ablation has been provided in the Figure. 6(b) of the main paper, where performance first increases and then decreases as $\beta$ becomes smaller. Accordingly, $\beta$=0.7 is adopted as default configuration. **Please refer to Sec. 3.2.3 of the main paper for detailed ablation analysis.**
>
> If you have any further questions or would like to discuss any aspect in more detail, please feel free to contact us at any time. We would be more than happy to provide additional clarification or engage in further discussion.

---

> ### Author Response · Authors · 2025-11-26
> **Looking Forward to Your Feedback and Any Further Comments**
>
> Dear Reviewer n5TF,
>
> Thanks very much for your thoughtful feedback on our submission. We have carefully considered your comments and updated our responses accordingly.
>
> We absolutely do not mean to rush you. Rather, as the discussion period is approaching its end, we want to kindly follow up and check whether you might have any additional comments or concerns that we could further address. If possible, we would be very grateful to know whether our responses have adequately addressed your comments. Your feedback is invaluable to us, and we would be pleased to provide any further clarification you might need.
>
> Thank you again for your time and consideration.
>
> With our best regards,
>
> The Authors of Submission 12662

---

### Author Response · Authors · 2025-12-03
**Summary of the Discussion for the Area Chair**

Dear Area Chair,

We sincerely thank all the reviewers for their constructive comments. We also appreciate your time and effort in examining both the reviewers’ assessments and our responses to reach a fair and well-informed decision. To facilitate your evaluation, we summarize the key points of our responses to the reviewers’ comments and reviewers’ feedback.

Overall:

The origin scores are
+ **8 (confidence: 5)**
+ **8 (confidence: 4)**
+ **6 (confidence: 2)**
+ **4 (confidence: 4)**

**Most reviewers gave positive assessments (avg. score 6.5, avg. confidence 3.75)**. All reviewers’ concerns and questions have been addressed in point-by-point responses. Unfortunately, there is no feedback from the reviewers, though we sent messages to remind them of feedback.

### For the Reviewer EExj (rating: 8):

The reviewer highly praised the novelty and interest of our method, its strong empirical performance, and the model-agnostic compatibility of the framework, and gave good or excellent scores in soundness, presentation, and contribution.

In our response, we analyzed where the performance gains come from for both continuous-valued and discrete-valued ordinal regression tasks. We added experiments on medical datasets to further demonstrate GoR’s strong generalization across domains. We also supplemented the discussion of CLIP-based methods and pointed out where the reviewer-mentioned comparisons appear in the main paper. Finally, we provided an extended discussion on the potential and feasibility of deploying continuous generative approaches.

### For the Reviewer yYd1 (rating: 8):

This reviewer highly praised the novelty and theoretical soundness of our method, its strong empirical performance and generalization across domains, and its potential for future work.

In our response, we explained how GoR achieves reasonable sequence length through systematic vocabulary design and label decomposition, supplemented efficiency evaluations, and pointed to real deployment evidence demonstrating that GoR meets industrial latency constraints and achieves a practical latency-performance balance to address the reviewer’s concerns on efficiency. We further showed from the perspectives of reasonable sequence length and effective optimization how GoR mitigates error accumulation and exposure bias, and through decoding-length ablation demonstrated that the generative paradigm itself is a key contributor to its strong predictive performance.

### For the Reviewer BLVB (rating: 6):

This reviewer praised the novelty, rigor, effectiveness, and generalizability of GoR, and viewed it as a promising unified paradigm for ordinal modeling. The reviewer also gave good scores in soundness, presentation, and contribution.

In our response, we referenced the existing experiments and added supplementary results to show that the gains come from the paradigm itself rather than a stronger model architecture (Q1). We explained how GoR achieves reasonable sequence length and strong cross-domain generalization through systematic vocabulary design and label decomposition, without manual tuning (Q2 & Q5). For efficiency concerns (Q3), we pointed to real deployment evidence demonstrating that GoR satisfies industrial latency constraints and achieves a practical latency-performance balance. For compounding errors or exposure bias (Q4), we highlighted that GoR seamlessly integrates with existing generative optimization methods and pointed to the corresponding optimization experiments in the main paper.

### For the Reviewer n5TF (rating: 4):

Although this reviewer was the only one to give a negative rating of 4, the comments remained overall positive and focused on technical details rather than methodological concerns. The reviewer also highlighted that our method is well-written, easy to integrate into existing generative methods, and achieves strong results across 17 datasets in 6 domains. The questions are mainly about pruning effects, sequence-length control, and hyperparameter selection.

In our response, we provided detailed explanations of the vocabulary construction and CoDi pruning mechanism to show how they promote more balanced token usage and act as beneficial redundancy reduction that improves generalization. Additional experiments were included to demonstrate the efficiency-effectiveness trade-off under different length controls, and we provided real deployment evidence showing that GoR satisfies industrial latency constraints and achieves a practical latency–performance balance. For hyperparameters, we pointed to the relevant ablation studies and supplied additional experiments.

Unfortunately, the reviewer didn’t provide any feedback during the discussion period despite our reminder. From our side, all points in the original review have been addressed.

We have incorporated the key points and additional experiments into the revision, which we believe fully address the reviewers’ concerns.

Thanks again for your time and effort.

---

### Meta-Review · Area_Chair_quCj · 2025-12-25

**Summary:**

This paper received ratings of 4, 6, 8, 8. The reviewers appreciated the novelty, effectiveness, flexibility, practicality, theoretical justification and expensive experiments across diverse domains. Their questions include unclear contributions between the stronger decoder and the paradigm, latency, compounding errors from autoregressive recoding, parameter setting, and interface time. The authors provided detailed responses with additional experiments. No reviewers replied to the reviewers’ responses during the shorten rebuttal period. The AC reads the reviewers’ comments and authors rebuttal and believes that the reviewers’ concerns have needed well-addressed. The AC recommends accepting this paper.

**Reviewer Concerns:**

The authors addressed all the concerns of the reviewers.

**Reviewer Scores:**

Reviewers EExj, yYd1 and BLVB unlikely change score because they gave high ratings before the start of the rebuttal period.
Reviewer n5TF likely changes score since he is only one giving negative score, and the AC believes that the authors have addressed his concerns.

---

### Decision · Program_Chairs · 2026-01-26

Accept (Poster)